# Rice Secondary Metabolites: Structures, Roles, Biosynthesis, and Metabolic Regulation

**DOI:** 10.3390/molecules23123098

**Published:** 2018-11-27

**Authors:** Weixuan Wang, Yuying Li, Pengqin Dang, Siji Zhao, Daowan Lai, Ligang Zhou

**Affiliations:** Department of Plant Pathology, College of Plant Protection, China Agricultural University, Beijing 100193, China; wwxcau@163.com (W.W.); yylimail@163.com (Y.L.); 17835422208@163.com (P.D.); zsj149@126.com (S.Z.); dwlai@cau.edu.cn (D.L.)

**Keywords:** rice, *Oryza sativa*, secondary metabolites, phytoalexins, biological functions, biosynthesis, elicitation, metabolic regulation

## Abstract

Rice (*Oryza sativa* L.) is an important food crop providing energy and nutrients for more than half of the world population. It produces vast amounts of secondary metabolites. At least 276 secondary metabolites from rice have been identified in the past 50 years. They mainly include phenolic acids, flavonoids, terpenoids, steroids, alkaloids, and their derivatives. These metabolites exhibit many physiological functions, such as regulatory effects on rice growth and development, disease-resistance promotion, anti-insect activity, and allelopathic effects, as well as various kinds of biological activities such as antimicrobial, antioxidant, cytotoxic, and anti-inflammatory properties. This review focuses on our knowledge of the structures, biological functions and activities, biosynthesis, and metabolic regulation of rice secondary metabolites. Some considerations about cheminformatics, metabolomics, genetic transformation, production, and applications related to the secondary metabolites from rice are also discussed.

## 1. Introduction

Rice (*Oryza sativa* L.), which belongs to the Gramineae family, has been consumed by humans for almost 5000 years. Rice is a widely diffuse staple food, providing energy and nutrients for more than half of the world population, especially in Asia [1]. The most common rice consumed by humans is white rice (about 85%), and the rest is pigmented rice [2]. Furthermore, rice is a model plant for molecular studies of monocotyledonous species [3]. There are diverse secondary metabolites produced in rice. These metabolites are organ- and tissue-specific. For example, diterpenoid phytoalexins are mainly present in the leaves, whereas phenolic acids, flavonoids, sterols, and triterpenoids are mainly present in the bran [4]. Rice secondary metabolites play roles either as defense agents, by providing disease resistance and exerting anti-nematodal, anti-insect, and allelopathic ativities against biotic and abiotic stresses, or as plant growth regulators. They also show various kinds of biological activities, such as antimicrobial, antioxidant, cytotoxic, and anti-inflammatory properties, which are implicated in various health-promoting and disease-preventive effects. Rice metabolites mainly include phenolic acids, flavonoids, terpenoids, steroids, alkaloids. Some metabolites such as phenolic acids and flavonoids are also distributed in other plant species [5,6]. To our knowledge, many reviews have discussed a specific topic of rice secondary metabolites [7,8,9,10,11,12,13,14], but no review has focused on describing the whole variety of secondary metabolites of rice so far. Furthermore, significant advances on rice secondary metabolism have been made recently based on genomic, biosynthesis regulation, and metabolomic approaches [15,16,17,18]. In this review, we summarize and discuss the developments from studies on the structural diversity, biological functions, biosynthesis, and metabolic regulations of rice secondary metabolites.

## 2. Structural Diversity and Roles of Rice Secondary Metabolites

Rice can accumulate a large number of secondary metabolites, such as phenolic acids, flavonoids, terpenoids, steroids, and alkaloids. These molecules play various physiological and ecological roles (i.e., antimicrobial, insecticidal, growth regulatory, and allelopathic activities). They also exhibit features beneficial to humans, including cytotoxic, anti-tumor, anti-inflammatory, antioxidant, and neuroprotective properties. For example, many phenolic acids, flavonoids, tocopherols, tocotrienols, γ-oryzanol, and phytic acid from rice exhibit antioxidant activities [10].

### 2.1. Phenolic Acids and Their Biological Functions

Rice phenolic acids can be classified as soluble-free, soluble-conjugated, and insoluble-bound forms. The insoluble-bound phenolic acids covalently boind to structural components of cells like cellulose, hemicellulose, lignin, pectin, rod-shaped structural proteins, etc. [19]. The distribution of rice phenolic acids exhibits varietal differences, and rice bran has the highest total phenolic acid content among four different fractions of whole rice grain [20,21]. Overall, *p*-hydroxybenzoic acid (**2**), caffeic acid (**7**), protocatechuic acid (**10**), ferulic acid (**17**/**19**), sinapic acid (**27**), syringic acid (**30**), and vanillic acid (**32**) are present in the whole rice grain, and ferulic acid (**17**/**19**) is the most abundant phenolic acid in the insoluble-bound fraction [22]. Normally, the pigmented rice contains phenolic acids with a larger structural diversity and in higher content than the non-pigmented rice [2,13]. About 32 phenolic acid analogues have be identified in rice. Rice phenolic acids and their biological activities are listed in Table 1. The structures of rice phenolic acids are shown in Figure 1. Most rice phenolic acids have antioxidant activities, though some of them have not been evaluated individually but only mixed with other rice phenolic acids [20].

Zaupa et al. revealed that the main rice phenolic acids are protecatechuic acid (**10**), *p*-coumaric acid (**15**), ferulic acid (**17**/**19**), sinapic acid (**27**), and vanillic acid (**32**) [23]. Ding et al. investigated eight rice varieties of *O. sativa* sp. *japonica* and *O. sativa* sp. *indica* planted in different areas of China for their phenolic acids distribution by using UPLC-MS method. A total of 12 phenolic compounds were identified in all rice varieties. Protocatechuic acid (**10**), ferulic acid (**17**/**19**), gallic acid (**24**), and syringic acid (**30**) were the dominant phenolic compounds in rice bran, while *p*-hydroxybenzaldehyde (**1**) was the main phenolic acid in rice husk. Bran and husk fractions provide more than 90% of phenolic acids and antioxidant activity of the whole rice plant. In addition, the rice subspecies *japonica* has significant higher phenolic acids content and antioxidant activity than the *indica* subspecies [24]. Ferulic acid (**17**/**19**) has also been found as the major phenolic compound in black rice bran, indicating the potential use of black rice bran as a natural source of antioxidants [25].

Phenolic acids are considered to be natural antioxidants, being able to scavenge free radicals that may increase oxidative stress and potentially damage large biological molecules such as lipids, proteins, and nucleic acids [26]. Therefore, the phenolic acid content was positively correlated with rice antioxidant capacity [27]. The development and utilization of phenolic acid analogues from rice husk and bran are important for improving the functionality of rice by-products.

Some phenolic acids are released from rice roots as allelochemicals. The main phenolic acids in the root exudates were identified as *p*-hydroxybenzoic (**2**), caffeic (**7**), *p*-coumaric (**15**), syringic (**30**), and vanillic (**32**) acids [28]. As these phenolic acids are released at relatively low concentrations in the soil and other plant species have a high level of tolerance against phenolic acids, they are considered the least important allelochemicals in rice [8,29,30].

### 2.2. Flavonoids and Their Biological Functions

According to the structural features, rice flavonoids can be classified as flavones (**33**~**65**), flavonols (**66**~**77**), flavanones (or dihydroflavones, **78**~**83**), flavanonols (**84**~**87**), flavanols (**88**, **89**), and anthocyanins (**90**~**101**), along with their glycosides. Rice flavonoids mainly have antioxidant properties, though some of them have not been evaluated for their antioxidant activities [10]. Among them, anthocyanins are mainly distributed in pigmented rice plants [2]. Rice flavonoids and their biological activities are listed in Table 2. Their structures are shown in Figure 2.

Two apigenin *C*-glycosides schaftoside (**45**) and isoschaftoside (**46**) were identified in whole rice leaves [35] and phloem [36]. The contents of both flavones were higher in the phloem of an insect-resistant rice variety than in a susceptible variety, which suggested that schaftoside (**45**) and isoschaftoside (**46**) in rice act as an antifeedant against brown planthopper (*Nilaparvata lugens*) [37].

Flavones **56**~**64** belong to flavonolignans. Both tricin 4′-*O*-(*erythro*-β-guaiacylglyceryl) ether (**57**) and tricin 4′-*O*-(*threo*-β-guaiacylglyceryl) ether (**58**) from Njavara rice bran had cytotoxic activity and induced apoptosis in multiple tumor cells by themitochondrial pathway, which indicated their possible role as potential cytotoxic agents against cancer cells [38].

The flavonoids in rice include aglycones (i.e., quercetin, kaempferol and tricin) and their glycosides. Eight flavonoids, i.e., brassicin (**66**), isorhamnetin-4′-*O*-β-d-glucopyranoside (**67**), brassicin-4′-*O*-β-d-glucopyranoside (**68**), isorhamnetin-7-*O*-β-d-cellobioside (**69**), 3′-*O*-methyltaxifolin (**84**), 3′-*O*-methyltaxifolin-7-*O*-β-d-glucopyranoside (**85**), 3′-*O*-methyltaxifolin-4′-*O*-β-d-glucopyranoside (**86**), and 3′-*O*-methyltaxifolin-5-*O*-β-d-glucopyranoside (**87**), were isolated from *Oryza sativa* sp. *japonica* c.v. Hwa-Young. This cultivar has a high flavonoid content in the seeds and, particularly, in the endosperm tissue [39].

Two flavones, i.e., *O*-glycosides 5,4′-dihydroxy-3′,5′-dimethoxy-7-*O*-β-glucopyranosylflavone (**37**) and 7,4′-dihydroxy-3′,5′-dimethoxy-5-*O*-β-glucopyranosylflavone (**38**), were identified in allelopathic rice seedlings. Only their aglycone, 5,7,4′-trihydroxy-3′,5′-dimethoxyflavone (**36**), was found in the soil. These two flavone *O*-glycosides were exuded from the rice roots to the rhizosphere and were then transformed into their aglycone forms, which showed an allelopathic effect on associated weeds and microbes [40,41].

Sakuranetin (**81**) is a flavanone-type phytoalexin in rice active against plant pathogens. Naringenin (**79**) is considered the biosynthetic precursor of sakuranetin (**81**) in rice. The bioconversion of naringenin (**79**) into sakuranetin (**81**) is catalyzed by naringenin 7-*O*-methyltransferase (OsNOMT) in rice leaves [42]. The antifungal activity of sakuranetin (**81**) was found to be higher than that of naringenin (**79**) [43,44]. Very interestingly, sakuranetin (**81**) can be detoxificated into naringenin (**79**) and sternbin (**83**) by the rice blast pathogen *Magnaporthe oryzae* [44] and can also be detoxificated into naringenin (**79**), naringenin 7-*O*-β-d-xylopyranoside (**80**), and sakuranetin 4′-*O*-β-d-xylopyranoside (**82**) by the rice sheath blight pathogen *Rhizoctonia solani* [45].

Sakuranetin (**81**) is not only a plant antibiotic but also a potential pharmaceutical agent that induces adipogenesis in 3T3-L1 cells through enhanced expression of peroxisome proliferator-activated receptor γ2, contributing to the maintenance of glucose homeostasis in animals [46] and exhibits anti-inflammatory activity by inhibiting 5-lipoxygenase, which is involved in arachidonic acid metabolism in animal cells [47], anti-mutagenic activity [48], anti-*Helicobacter pylori* activity by inhibiting β-hydroxyacylacyl carrier protein dehydration [49], and antileishmanial and antitrypanosomal activities [50]. Sakuranetin (**81**) strongly stimulated melanogenesis in B16BL6 melanoma cells via the ERK1/2 and PI3K–AKT signaling pathways, which led to the upregulation of *Tyr* family genes, *TRP1* and *TRP2* [51].

Anthocyanins are widely distributed in black rice. Eight anthocyanins, i.e., cyanidin (**90**), cyanidin 3-*O*-gentiobioside (**91**), cyanidin 3-*O*-glucoside (**92**), cyanidin 3-*O*-rutinoside (**93**), cyanidin 3-*O*-sambubioside (**94**), cyanidin 3,5-*O*-diglucoside (**95**), peonidin (**100**), and peonidin 3-*O*-glucoside (**101**) were identified from the kernels of black rice by UPLC-Q-TOF-MS [52]. They showed obviously antioxidant activities. The protective effects were mainly due to their free radical scavenging capacity [52].

### 2.3. Terpenoids and Their Biological Functions

Rice terpenoids include monoterpenoids, sesquiterpenoids, diterpenoids, and triterpenoids. Some monoterpenoids and sesquiterpenoids are volatile components and are often distributed in rice leaves. Rice diterpenoids play roles as phytohormones and phytoalexins. The triterpenoids are usually distributed in rice bran. The monoterpenoids, sesquiterpenoids, and triterpenoids usually play functions as allelochemicals.

#### 2.3.1. Monoterpenoids and Their Biological Functions

Monoterpenoids are mainly volatile compounds which confer rice its good aroma character. They can be extracted from the headspace of some rice bran samples by solid-phase microextraction (SPME). At least 18 monoterpenoids have been identified in rice. Their names and biological activities are listed in Table 3. Their structures are shown in Figure 3.

Rice monoterpenoids are synthesized by various types of terpene synthases (TPSs), such as OsTPS20 and OsTPS24. These TPSs contain a transit peptide for localization in the chloroplasts where monoterpenes are biosynthesized from geranyl diphosphate (GPP) by TPSs via the 2-*C*-methyl-d-erythritol 4-phosphate (MEP) pathway. TPSs can be induced by jasmonic acid (JA). The amount of γ-terpinene (**117**) increased after JA treatment. γ-Terpinene (**117**) had significant antibacterial activity against *Xoo*. However, it did not show significant antifungal activity against the rice blast pathogen. The antibacterial mechanism of γ-terpinene (**117**) against *Xoo* involved damage to bacterial cell membranes [62].

Monoterpenes (*S*)-limonene (**107**), myrcene (**111**), α-pinene (**113**), sabinene (**115**), α-terpinene (**116**), and α-thujene (**119**) were detected from one-week-old *Xoo*-infected rice seedlings by the method of solid-phase microextraction-GC-MS. However, only (*S*)-limonene (**107**) severely inhibited *Xoo* growth, which suggests that (*S*)-limonene (**107**) plays a significant role in suppressing *Xoo* growth in rice seedlings [63].

Many volatile monoterpenoids including linalool (**108**) were accumulated in response to the exogenous application of JA. The xpression of linalool synthase gene was upregulated by JA. Vapour treatment with linalool (**108**) induced resistance to *Xoo*. The transgenic rice plants overexpressing linalool synthase gene were more resistant to *Xoo*, which suggests that linalool (**108**) plays an important role in JA-induced resistance to *Xoo* [64].

#### 2.3.2. Sesquiterpenoids and Their Biological Functions

Sesquiterpenoids are also volatile components which contribute to the aroma quality of rice. They can be analyzed and identified by GC and GC-MS. The relative content of sesquiterpenoids was much lower, on average, than that of monoterpenoids in rice. Sesquiterpenoids are usually produced and released from wounds or microbe-infected sites. They act as signaling molecules that induce defense against tissue damage caused by herbivores or plant pathogens [67]. Rice sesquiterpenoids and their biological activities are listed in Table 4. Their structures are shown in Figure 4.

Sesquiterpenes are biosynthesized from farnesyl diphosphate (FPP) by TPSs via the mevalonate (MVA) pathway in the cytoplasm. Rice terpene synthase 18 was found to localize in the cytoplasm and synthesized the sesquiterpenes (*E*)-nerolidol (**139**) and (*E*)-β-farnesene (**132**), whose amounts increased after JA treatment. (*E*)-Nerolidol (**139**) had significant antibacterial activity against *Xoo* [68]. Rice sesquiterpenoids and their biological activities are listed in Table 4.

#### 2.3.3. Diterpenoids and Their Biological Functions

Almost all rice diterpenoids are members of the labdane-related superfamily, which includes not only phytohormone gibberellins (GAs) but also phytoalexins (i.e., phytocassanes, oryzalides, and oryzalexins), participate in the defense against pathogens, and are allelochemicals (i.e., momilactone B) inhibiting the growth of other plant species. Rice diterpenoids and their biological activities are listed in Table 5. Their structures are shown in Figure 5.

The major endogenous GA in rice was identified as GA_19_ (**147**). Other GA analogs are GA_1_ (**145**) and GA_4_ (**146**). The level of active Gas, such as GA_1_ (**145**), may be regulated by the rate of biosynthesis of GA_19_ (**147**) or its metabolic conversion [70].

Up to now, 37 diterpenoid-type phytoalexin analogues have been identified from rice plants. They have been further classified into five subtypes according to their biosynthetic pathways and structural characters [17]. The first one (**148**~**153**) is the pimaradiene type which mainly includes momilactones A (**148**) and B (**149**) [71] and 9β-pimara-7,15-diene-3β,6β,19-triol (**153**) [72]. The second subtype (**154**~**173**) is the *ent*-sandaracopimaradiene type which mainly includes oryzalexins A~F (**163**~**168**) [73,74,75,76,77,78,79]. The third one is the stemarene type that contains oryzalexin S (**174**) [44] and stemar-13-en-2α-ol (**175**) [72]. The fourth one is the *ent*-cassadiene type, containing phytocassanes A~F (**176**~**181**) [72,80,81,82]. The fifth one is the casbene type, including 5-deoxo-*ent*-10-oxodeprssin (**182**) [83], 5-dihydro-*ent*-10-oxodepressin (**183**) [83], and *ent*-10-oxodepressin (**184**) [84].

A few oryzalide-related compounds were isolated from the leaves of a cultivar resistant to the *Xoo*. They were identified as *ent*-15,16-epoxy-2,3-dihydroxy-kaurane (**154**) [85], *ent*-2,3,15-trihydroxy-kaurane (**155**) [85], *ent*-15,16-epoxy-kauran-3-one (**156**) [85], oryzadione (**157**) [86], *ent*-15,16-epoxy-3β-hydroxy-kauran-2-one (**158**) [86], *ent*-15,16-epoxy-3-oxa-kauran-2-one (**159**) [86], *ent*-15,16-epoxy-3β-myristoyloxy-kauran-2-one (**160**) [86], *ent*-15,16-epoxy-3α-palmitoyloxy-kauran-2-one (**161**) [86], *ent*-15,16-epoxy-3β-palmitoyloxy-kauran-2-one (**162**) [86], oryzalide A (**163**) [87,88], oryzalide B (**164**) [88], oryzalic acid A (**169**) [88], and oryzalic acid B (**170**) [85]. In contrast to typical diterpene phytoalexins, the accumulation of oryzalide-related comounds is only moderately induced by *Xoo* infection [89].

Three compounds, i.e., 9β-pimara-7,15-diene-3β,6β,19-triol (**153**), stemar-13-en-2α-ol (**175**), and phytocassane F (**181**) were accumulated following an infection by the rice blast pathogen *M. oryzae*. 9β-pimara-7,15-diene-3β,6β,19-triol (**153**) and stemar-13-en-2α-ol (**175**) exhibited weak antifungal activity and may be the biosynthetic intermediates of rice phytoalexins momilactones and oryzalexin S (**174**), respectively. Phytocassane F (**181**) exhibited relatively high inhibitory activity against the mycelial growth of *M. oryzae*, to the same extent as the known phytoalexin phytocassane A (**176**) [72].

Some diterpenoids such as momilactones A (**148**) and B (**149**) have their obvious allelopathic effects. Momilactones A (**148**) and B (**149**) mainly distribute in rice husks, leaves, seedlings, and straw. They function as either rice defense systems against pathogens and insects or growth inhibitors in seed dormancy [90]. Both momilacontes A (**148**) and B (**149**) inhibited the growth of barnyard grass (*Echinochloa crus-galli*) and *Echinochloa colonum*, the most noxious weeds in rice field, at concentrations greater than 1 and 10 μM, respectively. Momilactone B (**149**) exhibited greater growth inhibitory activity than momilactone A (**148**) [91]. Momilactone B (**149**) was preferentially secreted from the rice roots into the neighboring environment over the entire life cycle at phytotoxic levels. Momilaconte B (**149**) seems to account for the majority of rice allelopathy, while momilactone A (**148**) accumulates to higher levels in the plant upon infection. Interestingly, both momilactones A (**148**) and B (**149**) inhibited root and shoot growth of rice seedlings only at concentrations greater than 100 μM and 300 μM, respectively. Therefore, the ability of momilactones A (**148**) and B (**149**) to suppress the growth of rice seedlings was much lower than their effect on *E. crus-galli* and *E. colonum*, with no visible damage to rice seedlings exerted by momilactones A (**148**) and B (**149**) at levels that were cytotoxic to other plant species [91]. Selective removal of the momilactones from the complex mixture of rice root exudates significantly reduced allelopathy, which demonstrated that momilactones served as allelochemicals [8,92].

#### 2.3.4. Triterpenoids and Their Biological Functions

Triterpenoids are usually distributed in rice bran. Eight hydroxylated triterpene alcohol ferulates (**188**~**190**, **193**, **194**, **196**, **197**, **200**) were isolated from rice bran. They showed moderate cytotoxic activity [98,99]. The seed coats (or bran) usually contain large amounts of bioactive metabolites. This was also observed for the seed coats of quinoa (*Chenopodium quinoa*), where there were various triterpenoids distributed. Quinoa triterpenoids showed antimicrobial and molluscicidal activities [100]. There are few reports about the physiological and ecological functions of rice triterpenoids. The aglycones of rice triterpenoids are citrostadienol (**185**), cycloartenol (**191**), cycloeucalenol (**198**), gramisterol (**201**), and lupeol (**205**). On the basis of the biosynthetic pathway, citrostadienol (**185**), cycloeucalenol (**198**), gramisterol (**201**), and their derivatives are considered nortriterpenoids. Rice triterpenoids and their biological activities are listed in Table 6. Their structures are shown in Figure 6.

γ-Oryzanol is a mixture of triterpene and sterol ferulates extracted from rice bran [101]. In addition to its antioxidant activity, γ-oryzanol is often associated with cholesterol-lowering, anti-inflammatory, anti-cancer, and anti-diabetic properties [102]. The mixture of triterpene alcohols and sterols, with its components such as cycloartenol (**191**) and 24-methylene cycloartanol (**195**) from rice bran, can lower postpradial hyperglyceimia in mice and humans [103].

The main triterpene ferulates are cycloartenol ferulate (**193**/**194**) and 24-methylenecycloartanol ferulate (**196**/**197**). Both cycloeucalenol *trans*-ferulate (**194**) and 24-methylenecycloartanol *cis*-ferulate (**196**) showed anti-inflammatory activity in mice with inflammation induced by 12-*O*-tetradecanoylphorbol-13-acetate [104].

### 2.4. Steroids and Their Biological Functions

Plant steroids, generally termed phytosterols, are integral components of the membrane lipid bilayer in plants. They regulate membrane fluidity, influencing membrane’s properties, functions, and structure. An increase in the accumulation of sterols, namely, campesterol (**209**), β-sitosterol (**225**), and stigmasterol (**241**) was observed in rice as seedlings matured. These molecules are considered to have a role in drought stress tolerance in rice [109]. Steroids are usually distributed in the rice bran [107,110]. To date, 37 steroids have been identified from rice plants. Their names and biological activities are listed in Table 7. Their structures are shown in Figure 7.

Sterol ferulates are the main components of γ-oryzanol, which is isolated from rice bran. The main sterol ferulates are campesterol *trans*-ferulate (**215**) and sitosterol *trans*-ferulate (**229**) [102].

Some sterylglycosides (**231**~**235**), such as mono-, di-, tri-, tetra-, and pentaglycosylsterols, have been isolated from rice bran. The sugar component is glucose, and the glucose units are linked by β1,4-bonds [111,112].

Stigmastanol-3β-*p*-butanoxy dihydrocoumaroate (**238**) and stigmastanol-3β-*p*-glyceroxy dihydrocoumaroate (**239**) were isolated from rice hulls. Of them, stigmastanol-3β-*p*-butanoxy dihydrocoumaroate (**238**) showed weak growth inhibitory activity toward duckweed (*Lemna pausicostata*) [108].

### 2.5. Alkaloids and Their Biological Functions

2-Acetyl-1-pyrroline (2AP, **248**) is an important nitrogen-containing aroma compound that gives aromatic rice its characteristic flavor [113]. The concentration of 2AP (**248**) in uncooked Khao Dawk Mali 105 brown rice was quantitatively analyzed by capillary GC and found to be 0.34 μg/g [114]. This compound also occurs naturally in some other plants such as *Pandanus amaryllifolius* leaves and *Vallaris glabra* flowers [14]. Proline was proved to be the precursor for the biosynthesis of 2AP (**248**) in aromatic rice [115].

The main alkaloids in rice are phenylamides containing an indole ring. Rice plants accumulate phenylamides in response to a pathogen attack. If rice leaves are infected with the pathogens *Cochliobolus miyabeanus* and *Xanthomonas oryzae*, phenylamides are induced. They include *N*-feruloylagmatine (FerAgm, **244**), *N*-feruloylputrescine (FerPut, **245**), *N*-benzoylserotonin (BenSer, **249**), *N*-benzoytryptamine (BenTry, **250**), *N*-benzoyltyramine (BenTyr, **251**), *N*-*trans*-cinnamoylserotonin (CinSer, **252**), *N*-*trans*-cinnamoyltryptamine (**253**), *N*-*trans*-cinnamoyltyramine (CinTyr, **254**), *N*-*p*-coumaroylserotonin (CouSer, **255**), and *N*-feruloylserotonin (FerSer, **256**). Some of these phenylamides displayed antimicrobial activity against *C. miyabeaunus* and *X. oryzae*, indicating that they are phytoalexins [116]. Rice alkaloids and their biological activities are listed in Table 8. Their structures are shown in Figure 8.

### 2.6. Other Metabolites

Other secondary metabolites in rice include anthracenes (**263**~**265**), tocopherols (**269**~**272**), and tocotrienols (**273**~**276**). Their names and biological activities are listed in Table 9. Their structures are shown in Figure 9.

(*E*,*E*)-2,4-Heptadienal (**261**) is a JA-responsive volatile component in rice plants. (*E*,*E*)-2,4-Heptadienal (**261**) has both antibacterial and antifungal activities against *Xoo* and *M. oryzae*. In addition, it is also toxic to rice plants. (*E*,*E*)-2,4-Heptadienal (**261**) is essential for rice survival against pathogen attacks [120].

Three anthracene derivatives, i.e., orizaanthracenol (1-methoxyanthracen-2-ol, **263**), 1-hydroxy-7-((2*S*,3*R*,4*R*,5*S*)-2″,3″,4″-trihydroxy-5″-(hydroxymethyl)tetrahydro-2*H*-pyran-1-yloxy)anthracen-2-yl 3′,7′-dimethyloctanoate (**264**), and 1-hydroxy-7-((2*S*,3*R*,4*R*,5*S*)-2″,3″,4″-trihydroxy-5″-(hydroxymethyl)tetrahydro-2*H*-pyran-1-yloxy)anthracen-2-yl 3′,7′,11′,15′,19′-pentamethyltricosanoate (**265**), have been isolated from the rice hulls of *O. sativa*. Among the three compounds, orizaanthracenol (**263**) exhibited the highest inhibitory activity with respect to the germination of radish (*Raphanus sativus*) seeds, at 40 μg/mL [121].

(*Z*)-3-Hexen-1-ol (**262**) and other volatiles are released from elicitors (CuCl_2_, JA, UV, Met, and chitosan oligosaccharide)-treated and rice blast fungus-infested rice leaves [66].

(5*S*)-5-(Acetyloxy)-3-(1-methylenthyl)-2-cyclohexen-1-one (also named 3-isopropyl-5-acetoxycyclohexene-2-one-1 (**266**) is released from rice seedlings. It inhibited the growth of weeds *E. crus-galli* and *Cyperus difformis* [55].

*cis*-12-oxo-Phytodienoic acid (**267**) stimulated rice defense response to the brown planthopper (*Nilaparvata lungens*), a piercing-sucking insect pest of rice. This compound also stimulated the resistance of radish (*R. sativus*) seedlings to green peach aphid *Myzus persicai* which indicates the potential application of *cis*-12-oxo-phytodienoic acid (**267**) to stimulate plant defense responses to piercing-sucking insect pests in agriculture [122].

## 3. Biosynthetic Pathways of Rice Secondary Metabolites

In the 1980s, knowledge about the structures of rice secondary metabolites (i.e., phytoalexins) and their biosynthetic pathways accumulated, but no rice secondary metabolite biosynthetic enzyme genes was identified. In 2002, the draft sequences of rice genomes were published for *japonica* subspecies [15] and *indica* subspecies [16]. The annotation database platform for the rice genome was also developed and became publicly available (http://www.dna.affrc.go.jp/genome/#ricegenome) [9]. The biosynthetic genes for rice secondary metabolites are usually clustered in the genomes. The elucidation of their biosynthetic pathways is benefitting from the sequence of the rice genome, and many progresses have been achieved.

### 3.1. Biosynthesis of Flavonoids

A rice flavonoid biosynthesis pathway has been suggested by several studies, and the identified genes and enzymes involved in the pathway are shown in Figure 10 [124].

The biosynthesis of sakuranetin (**81**) has been given particular attention as this metabolite is an important phytoalexin in rice. In addition to the phytopathogenic infection induction [125], sakuranetin (**81**) can be induced by ultraviolet (UV) irradiation [43], treatment with CuCl_2_ [126], JA [127], methionine [128], the herbicides pretilachlor and butachlor [129], the bacterial phytotoxin coronatine [130], the phytopathogenic stem nematode *Ditylenchus angustus* [131], and the insect pest white-backed planthopper (*Sogatella furcifera*) [132].

Sakuranetin (**81**) has been revealed to be biosynthesized from naringenin by *S*-adenosyl-l-methionine-dependent naringenin 7-*O*-methyltransferase (NOMT), which is a key enzyme for sakuranetin production. NOMT was successfully purified and identified [42]. As naringenin (**79**) is a biosynthetic intermediate for a variety of flavonoids, NOMT plays a key role in sakuranetin biosynthesis at a branch point in the common flavonoid biosynthetic pathway (Figure 11). OsMYC2, which is an essential factor for JA-induced sakuranetin production in rice, interacts with MYC2-like proteins that enhance its transactivation ability [133]. The biosynthetic pathway of sakuranetin (**81**) is shown in Figure 11 [9].

### 3.2. Biosynthesis of Terpenoids

The biosynthesis of rice diterpenoid phytoalexins has been relatively detailed studied. The biosynthesis of the other terpenoids such as abscisic acid (ABA) and GAs in plants was discussed in reviews published elsewhere [134,135]. The production of rice diterpenoid phytoalexins can be induced by a series of stresses. For examples, phytocassanes can be induced by ultraviolet (UV) irradiation [72], and momilactone A (**148**) can be induced by the bacterial phytotoxins coronatine [130] and methionine [128].

The biosynthetic genes of diterpenoids are organized on the chromosome in functional gene clusters, comprising diterpene cyclase, dehydrogenase, and cytochrome P450 monooxygenase genes. Most of them in the rice genome are present in two gene clusters on chromosomes 2 and 4 (termed the Os02g cluster and the Os04g cluster, respectively) [136,137,138]. Their functions have been studied extensively using in vitro enzyme assay systems. Specifically, P450 genes (*CYP71Z6*, *Z7*, *CYP76M5*, *M6*, *M7*, *M8*) on rice chromosome 2 have multifunctional activities associated with *ent*-copalyl diphosphate-related diterpene hydrocarbons. Rice diterpenoids mainly contain phytohormones (i.e., gibberellins), phytoallexins (i.e., oryzalexins and phytocassanes), and allelochemicals (i.e., momilactones). They are biosynthesized via the MEP pathway in rice plants and have been well reviewed [139,140,141,142]. Gibberellins are considered phytohormones and are a large family of diterpenoids that possess the tetracyclic *ent*-gibberellane carbon skeletal structure arranged in either four or five ring systems, where the variable fifth ring is a lactone. Gibberellin biosynthesis and metabolism were well reviewed [143].

Rice diterpenoids are labdane-related. Their biosynthetic pathways in cultivated rice *O. sativa* are shown in Figure 12 [140,141].

Bioactive rice diterpenoids are commonly elaborated by the addition of at least two spatially separated hydroxyl groups. For example, orzyalexin D (**166**) is simply 3α,7β-dihydroxylated *ent*-sandaracopimaradiene, while orzyalexin E (**167**) is the 3α,9β-dihydroxy derivative. Notably, the production of these phytoalexins appears to proceed via the initial C3α hydroxylation of *ent*-sandaracopimaradiene catalyzed by OsCYP701A8, while OsCYP76M6 and OsCYP76M8 catalyze the subsequent hydroxylation at C9β or C7β, leading to the production of oryzalexins D (**166**) and E (**167**), respectively (Figure 13). These final biosynthetic steps represent the first complete pathways in the production of rice diterpenoid phytoalexins [144].

### 3.3. Biosynthesis of Tocotrienol and Tocopherol

Tocotrienol (T3), an unsaturated form of vitamin E with three double bonds in its isoprenoid side chain, is present in high concentration especially in rice grain [145].

Both tocotrienol and tocopherol (TOC) are biosynthesized through the pathways of mevalonate and shikimate [146]. Their biosynthetic pathways are shown in Figure 14.

### 3.4. Biosynthesis of Alkaloids

Both 2AP (**248**) and tryptophan biosynthesis pathways have been studied in detail, as 2AP is an important rice aromatic compound [147], and the tryptophan pathway is involved in rice defense responses against pathogenic infection via serotonin (**259**) production [148].

The biosynthesis pathway of 2AP production in rice starts with proline being catabolized via putrescine into γ-guanidinobutyraldehyde (GABald), a substrate of betaine aldehyde dehydrogenase (BAD2). If BAD2 is present and functional, it is able to convert the majority of GABald to γ-aminobutyric acid (GABA), but if BAD2 is absent or non-functional, the majority of GABald is acetylated to form 2AP [149]. The biosynthesis pathway of rice 2AP is shown in Figure 15.

Tryptophan decarboxylase (TDC) transforms typtophan (Trp) into tryptamine (**260**), consequently increasing the metabolic flow of tryptophan derivatives into the production of indole-containing metabolites. If the expression cassette containing *OsTDC* is inserted into an expression plasmid vector containing *OASA1D*, the overexpression of *OASA1D* significantly increases Trp levels in rice. The co-expression of *OsTDC* and *OASA1D* in rice cells led to almost complete depletion of the Trp pool and the consequent increase in the tryptamine pool. In recent years, the production of indole alkaloids has achieved great success through the metabolic engineering of the tryptophan pathway in rice [150]. The biosynthesis pathways of rice alkaloids are shown in Figure 16.

## 4. Metabolic Regulation of Secondary Meatobolites

The biosynthesis of plant secondary metabolites can be induced and regulated by various biotic and abiotic stresses, including organisms, jasmonic acid, oligosaccharides, and metal ions [9,151]. Among the secondary metabolites, phytoalexins are a kind of inducible antimicrobial metabolites whose biosynthesis is triggered not only by the invasion of pathogens including fungi, bacteria, and viruses, but also by a variety of abiotic elicitors, including phytohormones, oligosaccharides, UV irradiation, heavy metals (i.e., copper chloride), and mechanical stresses [152]. The main achievements regard the regulation of the biosynthesis of rice phytoalexins. The mechanisms of their biosynthetic regulation include signal recognition, signal transduction, gene expression, transcriptional and post-transcriptional pathways, and activation of the key enzymes.

### 4.1. Metabolic Regulation by Abiotic Stresses

#### 4.1.1. Metabolic Regulation by Phytohormones

Secondary metabolite biosynthesis can be mediated by phytohormones such as ABA), JA, cytokinins (CKs), salicylic acid (SA), ethylene (ET), and their conjugates [17,153].

JA, which is in the form of jasmonates, is a plant hormone which induces the biosynthesis of many secondary metabolites which play roles in plant-environment interactions [154]. JA can induce rice defense responses and plays an important role as a signal transducer for phytoalexin production in stress (e.g., CuCl_2_, oligosaccharides, phytotoxins)-stimulated rice leaves through gene activation. For example, the endogenous level of JA increased rapidly in CuCl_2_-stimulated rice leaves, and exogenously applied JA caused a large amount of phytoalexin production in rice leaves [126]. As far as we know, JA can induced the biosynthesis of momilactones A (**148**) and B (**149**) as well as of γ-terpinene (**117**) [62].

The basic leucine zipper transcription factor OsTGAP1 acts as a regulator of the coordinated production of diterpenoid phytoalexins in cultured rice cells. The inductive expression of *OstGAP1* under JA treatment was only observed in a root-specific manner, consistent with the JA-inducible expressions of the biosynthetic genes of diterpenoid phytoalexins in roots [155].

In addition, the amino acid conjugates of jasmonic acid *N*-[(−)-jasmonoyl]-*S*-isoleucine (JA-Ile) and *N*-[(−)-jasmonoyl]-*S*-phenylalanine were found to elicit the production of sakuranetin (**81**) in rice leaves. The elicitation was considered to arise from the induction of naringenin 7-*O*-methyltransferase, a key enzyme in sakuranetin biosynthesis [156]. Very interestingly, JA-Ile was not required for diterpenoid production in blast pathogen-infected or CuCl_2_-treated rice leaves [157].

The treatment with natural and synthetic CKs induced the production of diterpenoid phytoalexins in rice leaves and suspension-cultured cells [158]. However, CK treatment inhibited JA-inducible sakuranetin (**81**) production in rice leaves [127]. On the other hand, exogenous root applications of SA promoted the accumulation of oryzalexins and momilactone A (**148**) in the leaves [159]. A synergistic crosstalk of CK and SA signaling was also reported, showing that 0.1 mM CKs with benzothiadiazole (BTH), a plant activator that enhances SA signaling pathway, induced a several-fold enhancement of momilactone and phytocassane biosynthetic genes [160]. In addition, it was reported that the treatment of wounded rice leaves with methionine, the precursor of ET, induced the accumulation of sakuranetin (**81**) and momilactone A (**148**). Tiron, a free radical scavenger, counteracted the induction of both sakuranetin (**81**) and momilactone A (**148**) production in methionine-treated leaves, indicating that active oxygen species might be important in methionine-induced production of phytoalexins. However, ET treatment of wounded leaves induced the production of sakuranetin but not of momilactone A (**148**), suggesting that the induction of diterpenoid phytoalexin production by methionine was not regulated by ET alone [128]. In the susceptible rice cultivar Dorella, the bakanae pathogen (*Fusarium fujikuroi*) induced the production of gibberellin and abscisic acid and inhibited jasmonic acid production, and phytoalexin content in rice was very low [153].

In addition, ethylene-inducing xylanase from *Thichoderma viride* was a potent elicitor of immune responses in a variety of plant species, such as tobacco (*Nicotiana tabacum*), tomato (*Solanum lycopersicum*), and rice. This enzyme induced the expression of defense-related genes involved in the biosynthesis of phytoalexins (i.e., momilactones and phytocassanes) by a cation channel OsTPC1 in suspension-cultured rice cells [161].

#### 4.1.2. Metabolic Regulation by Oligosaccharides

Oligosaccharides derived from fungal and plant cell wall polysaccharides are a class of well characterized elicitors that can induce not only accumulation of secondary metabolites but also MAPK activation, ROS generation, and defense-related enzyme activities at a very low concentration, triggering plant innate immunity [162,163,164,165,166,167,168]. Studies were performed on the structure–activity relationships of the oligosaccharides, the characterization of the corresponding receptors, and the analysis of signal transduction cascades and elicitor-responsive genes. Early studies focused on the inducible effects of oligosaccharides on rice phytoalexin production. Their elicitation mechanisms have been partially revealed in recent years.

A mixture of chitin fragments obtained from the cell walls of the rice pathogen *Fusarium moniliforme* through either the action of constitutive rice chitinases or partial acid hydrolysis was active to induce diterpenoid phytoalexin biosynthesis in rice cell cultures in suspension [169]. *N*-Acetylchitooligosaccharides larger than hexaose induced the formation of momilactones A (**148**) and B (**149**) as well as oryzalexins A (**163**), B (**164**), and D (**166**) at concentrations of 10^−9^~10^−6^ M [170].

β-glucan fragments (oligosaccharides) from the cell walls of the rice blast fungus *M. oryzae* had the ability to elicit phytoalexin (i.e., momilactone A) biosynthesis in suspension-cultured rice cells. The potent elicitor glucopentaose, namely, tetraglucosyl glucitol, was purified from the digestion of the glucan by an endo-β-(1→3)-glucanase. Interestingly, the obtained tetraglucosyl glucitol from *M. oryzae* did not induce phytoalexin biosynthesis in the soybean cotyledon cells, indicating differences in the recognition of gluco-oligosaccharide elicitor signals in these two plants [171]. In addition, two purified oligosaccharide elicitors, *N*-acetylchitohepatose and tetraglucosyl glucitol, derived from *M. oryzae* cell walls, synergistically activated the biosynthesis of phytoalexin in suspension-cultured rice cells. Inhibition experiments for the binding of the radio labeled *N*-acetylchitooligosaccharide elicitor to the plasma membrane from rice cells indicated that the two elicitors are recognized by different receptors [172]. Rice cells recognize oligosaccharides for defense singnaling mainly through plasma membrane receptors [173,174].

Diterpenoid phytoalexins are synthesized through the plastidic MEP pathway in rice. OsTGAP1, a basic leucine zipper transcription factor, which is induced by the fungal chitin oligosaccharide elicitor, was identified as a key regulator of the coordinated expression of the clustered biosynthetic genes for diterpenoid phytoalexin production in rice [138]. The overexpression of the bZIP transcription factor OsbZIP79 resulted in the suppression of the chitin oligosaccharide-inducible expression of diterpenoid phytoalexin biosynthetic genes, and thus caused a decrease in the accumulation of diterpenoid phytoalexin in rice cells. OsbZIP79 is considered a negative regulator of rice diterpenoid phytoalexin production [175].

#### 4.1.3. Metabolic Regulation by Cerebrosides

Cerebrosides are categorized as glycosphingolipids. They are important components of a wide variety of tissues and organs in biological systems [176]. Cerebrosides were also found to occur in various fungi, such as *Cercospora solani-melogenae*, *Cochiliobolus miyabeansus*, *Fusarium oxysporum*, *Mycosperella pinodes*, *Rhizoctonia* sp., and *Trichoderma viride*, as the elicitors that can activate plant defense systems. They showed no antifungal activity against pathogens in vitro, showed phytoalexin-inducing activity when applied to plants by spray treatment, and also induced the expression of pathogenesis-related (PR) proteins in rice leaves [177].

Both cerebrosides A and C from the rice blast pathogen *M. oryzae* elicited hypersensitive cell death and phytoalexin (such as momilactone A and phytocassanes A and B) accumulation in rice plants [178].

The ceramides prepared from the cerebrosides by removal of glucose also showed elicitor activity even at lower concentations compared to the cerebrosides. In field experiments, the cerebroside elicitors effectively protected rice plants against the rice blast fungus. Cerebroside elicitors protected rice plants from other diseases as well and functioned as general elicitors in a wide variety of rice-pathogen interactions [179]. Further studies showed that cerebrosides are non-race-specific elicitors. Treatment of lettuce (*Lactuca sativa*), tomato (*Lycopersicon esculentum*), melon (*Cucumis melo*), and sweet potato (*Ipomoea batatas*) with cerebroside B resulted in resistance to infection of the fungal pathogens [180].

#### 4.1.4. Metabolic Regulation by Cholic Acid

Cholic acid (CA), a steroid elicitor of rice defense responses, was isolated from human feces. When rice leaves were treated with CA, defense responses were induced, with the accumulation of antimicrobial compounds, hypersensitive cell death, and pathogenesis-related (PR) protein synthesis. The induced antimicrobial compounds were identified as phytocassanes. The structure–activity relationship analysis showed that the hydroxyl groups at C-7 and C-12 and the carboxyl group at C-24 of cholic acid contributed to the elicitor activity [181]. In contrast to the other elicitors (i.e., fungal chitin oligosaccharide elicitor), cholic acid specifically triggered the accumulation of phytocassanes but not of momilactones, suggesting specificity in pathway regulation [180]. Further investigation of the effects of CA on the expression of diterpene cyclase genes showed that CA induced the transcription of the genes *OsCPS2* (*OsCyc2*) and *OsKSL7* (*OsDTC1*) involved in phytocassane biosynthesis. *OsCPS2* was particularly strongly induced, suggesting that it is one of the main mechanisms by which CA induces high levels of phytocassanes [182].

#### 4.1.5. Metabolic Regulation by Heavy Metal Ions

The role of heavy metal ions as antifungal agents may consists in part in inducing defense-response genes and in part in inhibiting the pathogens. Among the metal ions, copper ions (Cu^2+^) were the most effective to induce defense-related genes involved in phytoalexin biosynthesis [183]. Heavy metal ions are abiotic elicitors. Some metal ions can affect the production of plant secondary metabolites including phytoalexins [184]. Typical examples included tanshinone accumulation stimulated by metal ions Co^2+^, Ag^+^, and Cd^2+^ in *Salvia miltiorrhiza* cell cultures [185], andrographolide production elicitated by Cd^2+^, Ag^+^, Cu^2+^, and Hg^2+^ in *Andrographis paniculata* cell cultures [186], resveratrol production enhanced by Co^2+^, Ag^+^ and Cd^2+^ in *Vitis vinifera* cell cultures [187], and phaseollin production enhanced in Colombian bean (*Phaseolus vulgaris*) seedlings treated with CuCl_2_ [188].

The induction of phytoalexins by heavy metal ions in rice leaves was studied by punching detached leaves with a glass capillary tube and applying droplets of a heavy metal salt solution into the holes. Application of 1 mM copper chloride (CuCl_2_) induced the accumulation of oryzalexins A (**163**), B (**164**), C (**165**), and D (**166**) and of momilactones A (**148**) and B (**149**) in the tissues around the holes and in the droplets. Among the momilactones, momilactone A (**148**) showed a marked induction. Among the oryzalexins, oryzalexin B (**164**) was induced to the greatest extent. The accumulation of momilactone A (**148**) was first noted 12 hr after the application of copper ions, reaching a maximum after 72 h. As these phytoalexins accumulated, brown spots appeared in areas surrounding the punctured holes. Iron and mercury ions made up approximately 37% and 20% of the elicitor activity of copper ion (Cu^2+^), respectively. Manganese (Mn^2+^) and cobalt (Co^2+^) ions hardly showed any elicitor activity [189]. Further study showed that the endogenous level of JA increased rapidly in CuCl_2_-treated rice leaves. If rice leaves were treated with JA biosynthesis inhibitors, the production of phytoalexin elicited by CuCl_2_ decreased. JA is thus suggested to play an important role as a signaling molecule in CuCl_2_-elicited rice phytoalexin biosynthesis [126].

#### 4.1.6. Metabolic Regulation by Ultraviolet Irradiation

UV irraditon is very convenient for inducing the production of rice phytoalexins, because it is easy to irradiate many rice leaves with ultraviolet light [9].

An accumulation of oryzalexins A (**163**), B (**164**), C (**165**), and D (**166**), and of momilactones A (**148**) and B (**149**) accompanied by the appearance of brown spots on the leaf surface was observed in ultraviolet-irradiated detached rice leaves. Momilactone A (**148**) was detected in abundance, and among the oryzalexins, oryzalexin D (**166**) was a major substance. The content of these diterpenoid phytoalexins in rice leaves was dependent on leaf aging, the accumulation of these phytoalexins in the uppermost leaves being much lower than that in the aged leaves (lower leaves), and brown spots scarcely ever appeared on the surface of the uppermost leaves [190].

UV irradiation increased not only the concentration of momilactone B (**149**) in rice seedlings but also the secretion of momilactone B (**149**) into rice rhizosphere. As momilactone B (**149**) acts as an antimicrobial and allelopathic agent, the secretion of momilactone B into the rhizosphere may provide a competitive advantage for rice root establishment through the local suppression of soil microorganisms and the inhibition of the growth of competing plant species [191].

UV irradiation also stimulated flavonoid and phenylamide biosynthetic pathways in rice leaves. Five phytoalexins, i.e., sakuranetin (**81**), *N*-benzoyltryptamine (**250**), *N*-*trans*-cinnamoyltryptamine (**253**), *N*-*trans*-cinnamoyltyramine (**254**), and *N*-*p*-coumaroylserotonin (**255**) were isolated from UV-treated rice leaves [58]. In response to UV treatment, sakuranetin accumulated in rice leaves may serve as an antioxidant against UV-induced oxidative stress [42].

#### 4.1.7. Metabolic Regulation by Other Abiotic Stresses

Rice plants were treated with the fungicide 2,2-dichloro-3,3-dimethyl cyclopropanecarboxylic acid (WL28325). The biosynthesis of two phytoalexins, i.e., momilactones A (**148**) and B (**149**), was markedly enhanced. This suggested that the fungicide WL28325 has a systemic ability to activate host resistance against rice blast pathogens [192].

Pretilachlor and butachlor, two chloroacetamide herbicides that promote cell death, induced the accumulation of the phytoalexins momilactone A (**148**) and sakuranetin (**81**) in rice leaves. The accumulation of these phytoalexins was related to the herbicide concentration and the period of exposure and was followed by the appearance of necrotic lesions on the rice leaves [129].

Carbon dioxide (CO_2_) has the ability to regulate the biosynthesis of rice phenolics. With elevated CO_2_ concentrations, the total phenolic content as well as the DPPH radical scavenging capacity decreased, which indicated that these decreases may be meaningful in the preventive ability of rice against free radical-mediated degenerative diseases [193].

### 4.2. Metabolic Regulation by Biotic Stresses

#### 4.2.1. Metabolic Regulation by Bacteria

The biosynthesis of secondary metabolites can be regulated by either pathogenic and non-pathogenic bacteria or their extracts and components.

If rice leaves were infected with the bacterial pathogen *Xanthomonas oryzae*, phenylamides were induced. They were identified as *N*-feruloylagmatine (**244**), *N*-feruloylputrescine (**245**), *N*-benzoylserotonin (**249**), *N*-benzoytryptamine (**250**), *N*-benzoyltyramine (**251**), *N-trans*-cinnamoylserotonin (**252**), *N-trans*-cinnamoyltyramine (**254**), *N-p*-coumaroylserotonin (**255**), and *N*-feruloylserotonin (**256**) [116].

The bacterial phytotoxin coronatine was isolated from a *Pseudomonas syringae* pv. *atropurpurea* culture broth as a chlorosis-inducing compound in the leaves of Italian ryegrass (*Lolium multiflorum*). The structure of coronatine, an amide of coronafacic acid and coronamic acid, was somewhat related to that of JA. This phytotoxin could induce the accumulation of sakuranetin (**81**) and momilactone A (**148**) in rice leaves. Coronatine-inducible sakuranetin production is under the control of kinetin and ascorbic acid, as observed with JA. The similarity of the structures and elicitation manner of coronatine and JA suggests that they have a similar action mechanism leading to rice phytoalexin production [129].

#### 4.2.2. Metabolic Regulation by Fungi

The biosynthesis of rice secondary metabolites can be also regulated by fungi or their extracts and components. Fungal infection often results in the accumulation of phytoalexins in rice plants [194].

In the rice resistant cultivar Selenio, the presence of the pathogen *F. fujikuroi* induced a high production of sakuranetin (**81**), and symptoms of bakanae were not observed. On the contrary, in the susceptible genotype Dorella, the pathogen induced the production of gibberellin and abscisic acid and inhibited jasmonic acid production, and sakuranetin (**81**) content was very low [153]. Infection with the rice blast pathogen *M. oryzae* induced the production of momilactones, with greater accumulation of momilactones A (**148**) and B (**149**) [195], and also induced sakuranetin (**81**) production [125]. Young rice leaves in a resistant rice line exhibited a hypersensitive reaction (HR) within three days after the inoculation of a spore suspension of the blast pathogen *M. oryzae*, and an increase of sakuranetin was detected three days post-inoculation (dpi), increasing to four-fold at 4 dpi. In the susceptible line, increased sakuranetin was detected at 4 dpi but not at 3 dpi, by which time a large fungal mycelia had accumulated without HR. The decrease and detoxification of sakuranetin (**81**) were detected in both solid and liquid mycelium cultures of the rice blast pathogen [125].

Rice leaves accumulated serotonin (**259**) in response to infection by *Bipolaris oryzae*. If serotonin (**259**) was added to the culture media, it was converted into 5-hydroxyindole-3-acetic acid (5HIAA), which may be a detoxification process in the interaction between *B. oryzae* and rice [196].

When rice leaves were infected with *Cochliobolus miyabeanus*, phenylamides were also induced. They included *N*-feruloylagmatine (**244**), *N*-feruloylputrescine (**245**), *N*-benzoylserotonin (**249**), *N*-benzoytryptamine (**250**), *N*-benzoyltyramine (**251**), *N-trans*-cinnamoylserotonin (**252**), *N-trans*-cinnamoyltyramine (**254**), *N-p*-coumaroylserotonin (**255**), and *N*-feruloylserotonin (**256**) [116].

#### 4.2.3. Metabolic Regulation by Insect Pests

The attack by herbivorous insects is one of the major biological stresses that rice plants have to cope with. The secondary metabolites that are derived from the tryptophan pathway have been shown to play defensive roles against insects in rice plants. The biosynthesis of four tryptophan-derived metabolites was induced by the feeding on rice leaves of the rice striped stem-borer (*Chilo suppressalis)*. The amounts of *N-p*-coumaroylserotonin (**255**), *N*-feruloyltryptamine (**257**), serotonin (**259**), and tryptamine (**260**) in the larvae-fed leaves were 12-, 3.5-, 33-, and 140-fold larger than in the control leaves 48 h after the start of feeding [118].

The biosynthesis of serotonin (**259**) was suppressed by insect infestation in rice, which demonstrates that the regulation of serotonin biosynthesis plays an important role in the defense from insects. In rice, the cytochrome P450 gene *CYP71A1* encodes tryptamine 5-hydroxylase, which catalyses the conversion of tryptamine to serotonin. In susceptible wild-type rice, rice brown planthopper (*N. lugens*) feeding induces the biosynthesis of serotonin, whereas, in mutants with an inactivated *CYP71A1* gene, no serotonin is produced, and the plants are more insect-resistant [197].

When rice leaves were infested by the white-backed planthopper (*Sogatella furcifera*), accumulation of salicylic acid, jasmonic acid, and phytoalexins such as momilactone A (**148**) and sakuranetin (**81**) was observed. It is possible that *S. furcifera* released some elicitor compounds, which might be produced in its salivary glands, into the rice plants during feeding. Next, the defense signal systems, SA- and JA-mediated pathways, were activated by the elicitor. Finally, phytoalexins are induced in rice as antimicrobial compounds mainly through the activation of the JA-mediated pathway [132].

#### 4.2.4. Metabolic Regulation by Nematodes

The elicitation of phytoalexin synthesis was observed in rice in respononse to infection by nematodes. Both chlorogenic acid (**11**) and sakuranetin (**81**) were induced in the incompatible varieties of rice after infection by the stem nematode *Ditylenchus angustus*, and no change occurred in the susceptible varieties of rice [131]. In addition, the systemic suppression of metabolism in the shoot, including the isoprenoid and shikimate pathways, was observed upon rice infection by the root nematode *Hirschmanniella oryzae* [198].

#### 4.2.5. Metabolic Regulation by Viruses

The rice dwarf virus (RDV) P2 protein interacts with *ent*-kaurene oxidases, which play a key role in the biosynthesis of the growth hormones gibberellins in rice plants. This leads to reduced biosynthesis of gibberellins and to rice dwarf symptoms. In addition, the interaction between P2 protein and rice *ent*-kaurene oxidase-like proteins may decrease phytoalexin biosynthesis and make plants more competent for virus replication [199].

#### 4.2.6. Metabolic Regulation by Other Plants

Rice allelopathic activity increased in the presence of seedlings and root exudates of barnyard grass (*E. crus-galli*). This increase was not due to nutrient competition between the two plant species. Levels of momilactone B (**149**), both endogenous concentration in rice seedlings and secretion rate, were also increased by the presence of the seedlings and root exudates of barnyard grass. Probably, the active components from the root exudates triggered the production and secretion of momilactone B (**149**) [200]. Similarly, the production of the sorghum allelochemical sorgoleone was also induced by root extracts of the agriculturally relevant weed velvetleaf [201]. Accordingly, allelopathy potentially acts as an inducible defense mechanism mediated by the recognition of root exudate components specific to other plant species found in the relevant ecosystem [8]. The elicited compounds from rice root exudates need to be identified.

## 5. Conclusions and Future Perspectives

This review focuses on the elucidation of the structures, biological functions, biosynthesis, and metabolic regulation of rice secondary metabolites carried out during the past 50 years. Some metabolites (i.e., diterpenoid phytoalexins) and their metabolic pathways are unique to rice [141]. Some minor or new rice metabolites should be identified by using new techniques, such as chemoinformatics [202], metabolomics [7,18,34], and compound prediction based on biosynthetic gene clusters [9]. Furthermore, the biological activities of many isolated metabolites (i.e., phenolic acids, flavonoids, and terpenoids) need to be systematically evaluated.

The physiological functions of some rice secondary metabolites remain unknown. Commonly, the development of null mutants and transgenic over-expression lines is enabling the critical examination of biological functions. Phytoalexin-related gene over-expression generally results in increased resistance against pathogens in genetically modified plants [203,204].

The biosynthetic regulation of rice secondary metabolites is very complicated. Concentrated efforts have revealed the relevant factors and signaling pathways that are involved in the regulation of phytoalexin production in rice. It is still unknown, however, how transcription factors regulate phytoalexin biosynthetic genes in concert. Further research investigating the molecular mechanisms of the transcriptional regulation of phytoalexin biosynthetic genes and revealing how upstream signals activate each transcription factor in the signaling cascade is essential. Rice metabolic regulation is certain to far exceed the complexity of the biosynthetic pathways [11,17,205].

In addition, exogenous genes such as stilbene synthase (STS) gene were successfully transferrred from other plant species to rice to improve its resistance against rice blast disease [206]. The importatance of some secondary metabolites, such as 2-acetyl-1-pyrroline (**248**) [14], phytoalexins [204], and momilactones [207], has been emphasized in agriculture and in the medicinal and food industries. These will be important fields of application for rice secondary metabolites.

## Figures and Tables

**Figure 1 molecules-23-03098-f001:**
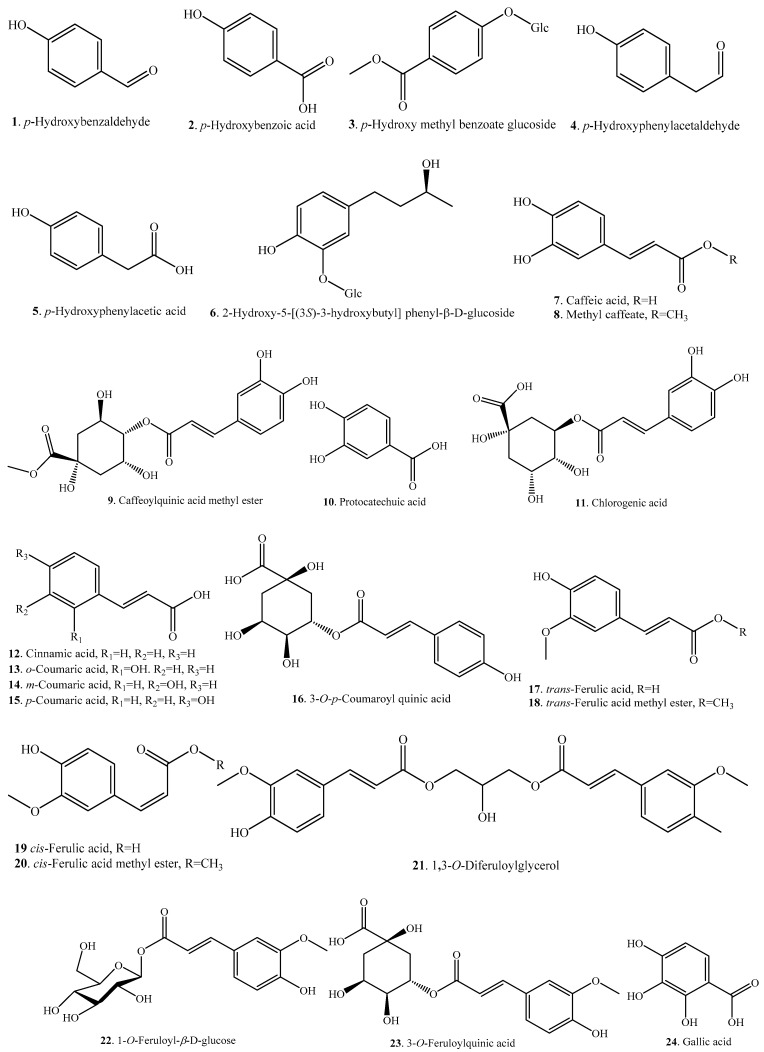
Structures of the phenolic acids isolated from rice.

**Figure 2 molecules-23-03098-f002:**
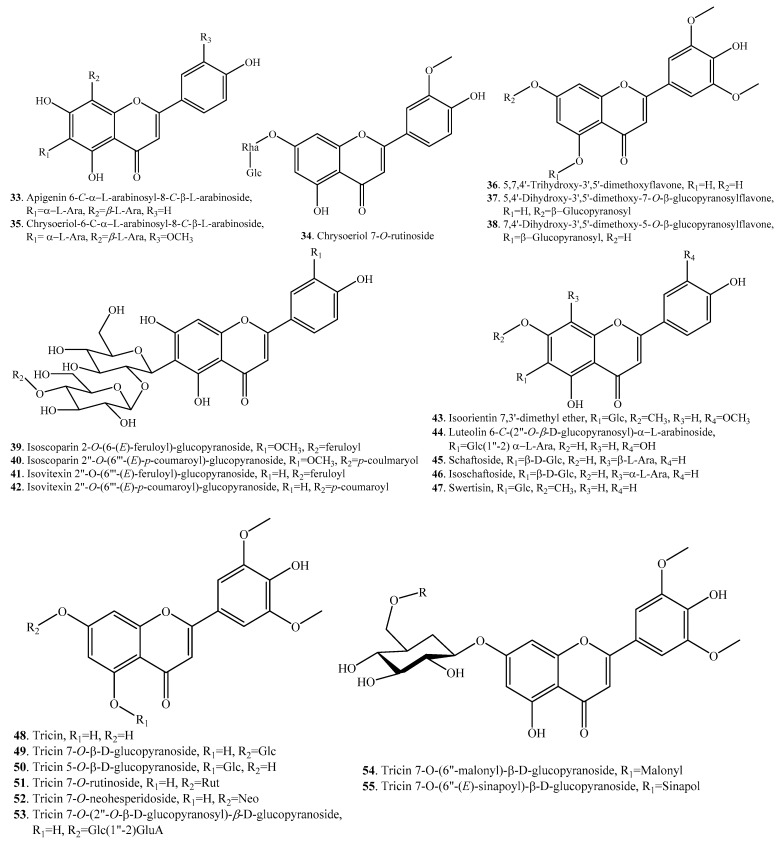
Structures of the flavonoids isolated from rice.

**Figure 3 molecules-23-03098-f003:**
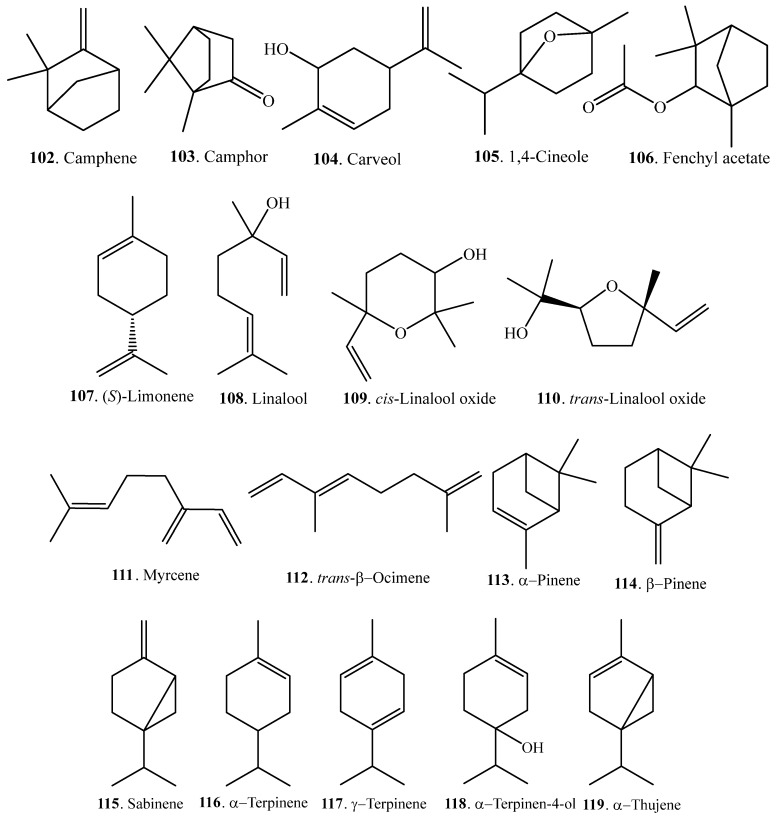
Structures of the monoterpenoids identified in rice.

**Figure 4 molecules-23-03098-f004:**
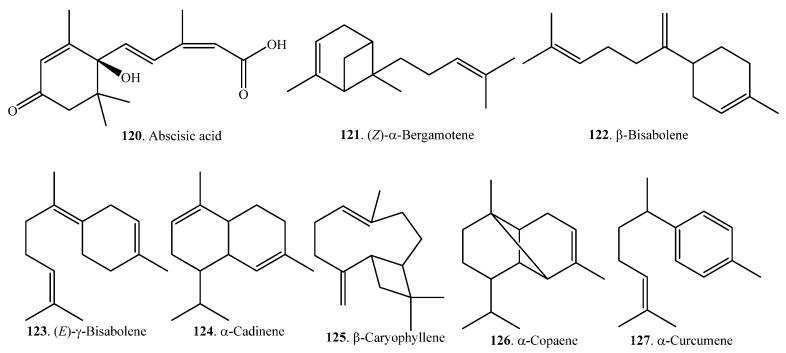
Structures of the sesquiterpenoids isolated from rice.

**Figure 5 molecules-23-03098-f005:**
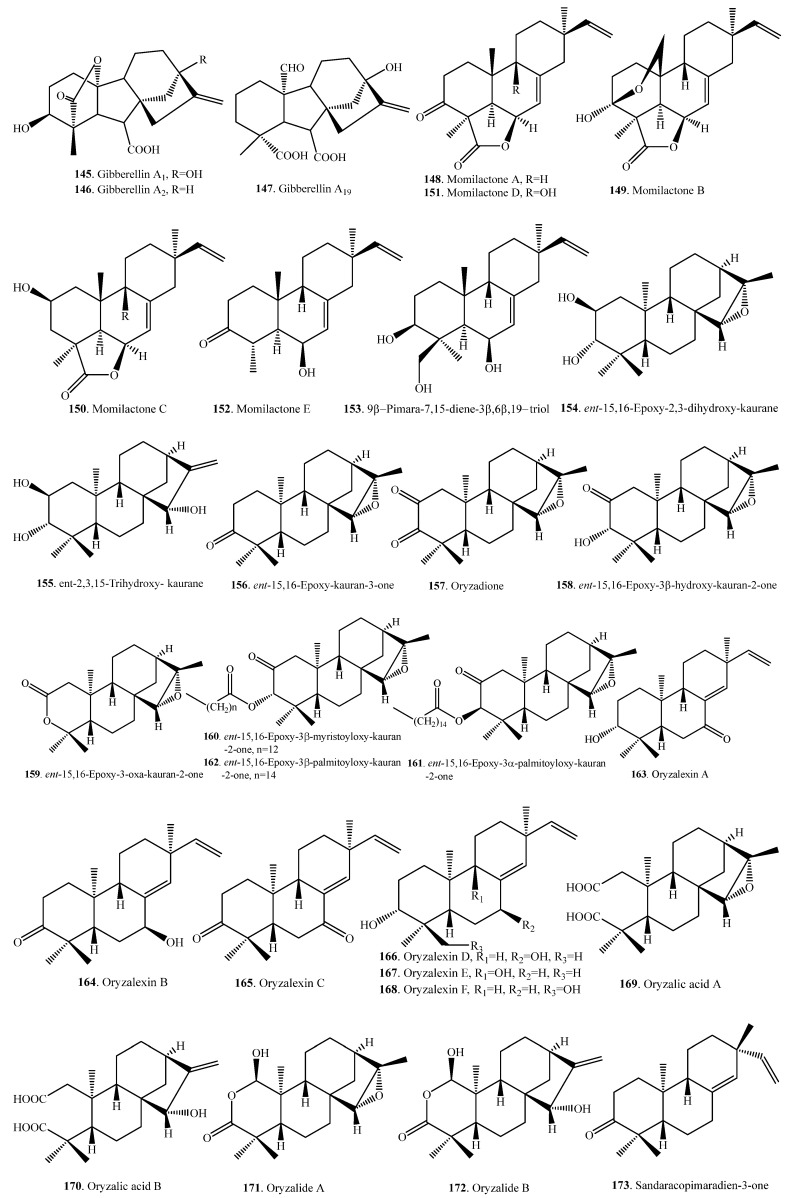
Structures of the diterpenoids isolated from rice.

**Figure 6 molecules-23-03098-f006:**
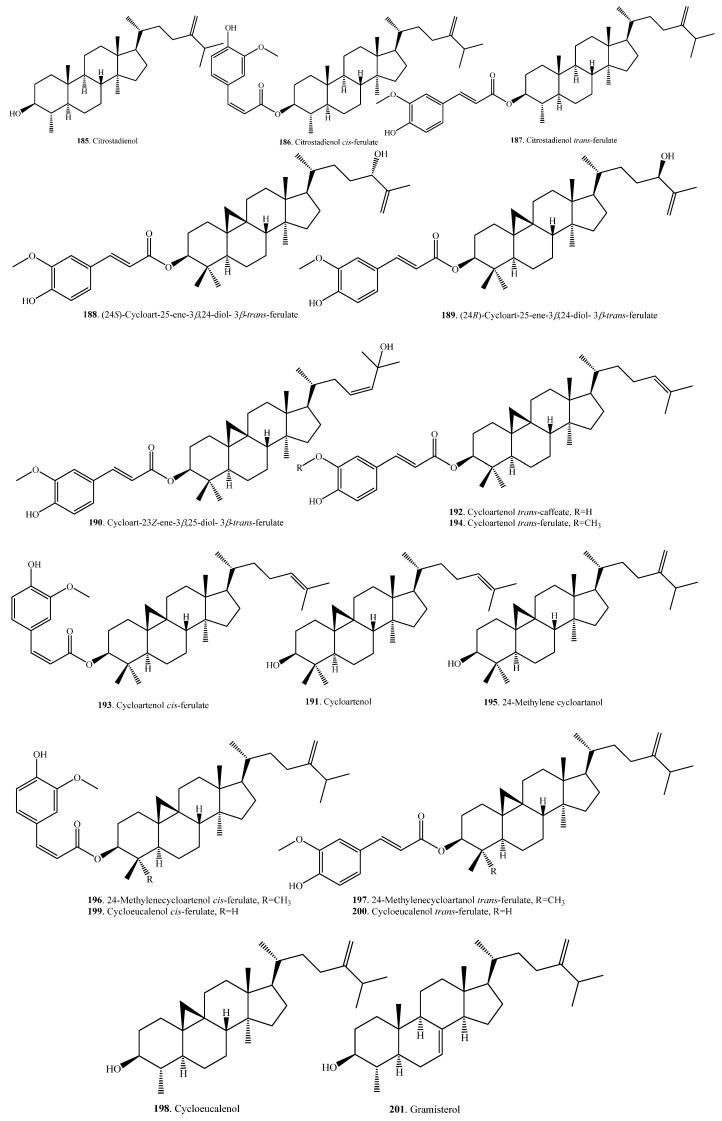
Structures of the triterpenoids isolated from rice.

**Figure 7 molecules-23-03098-f007:**
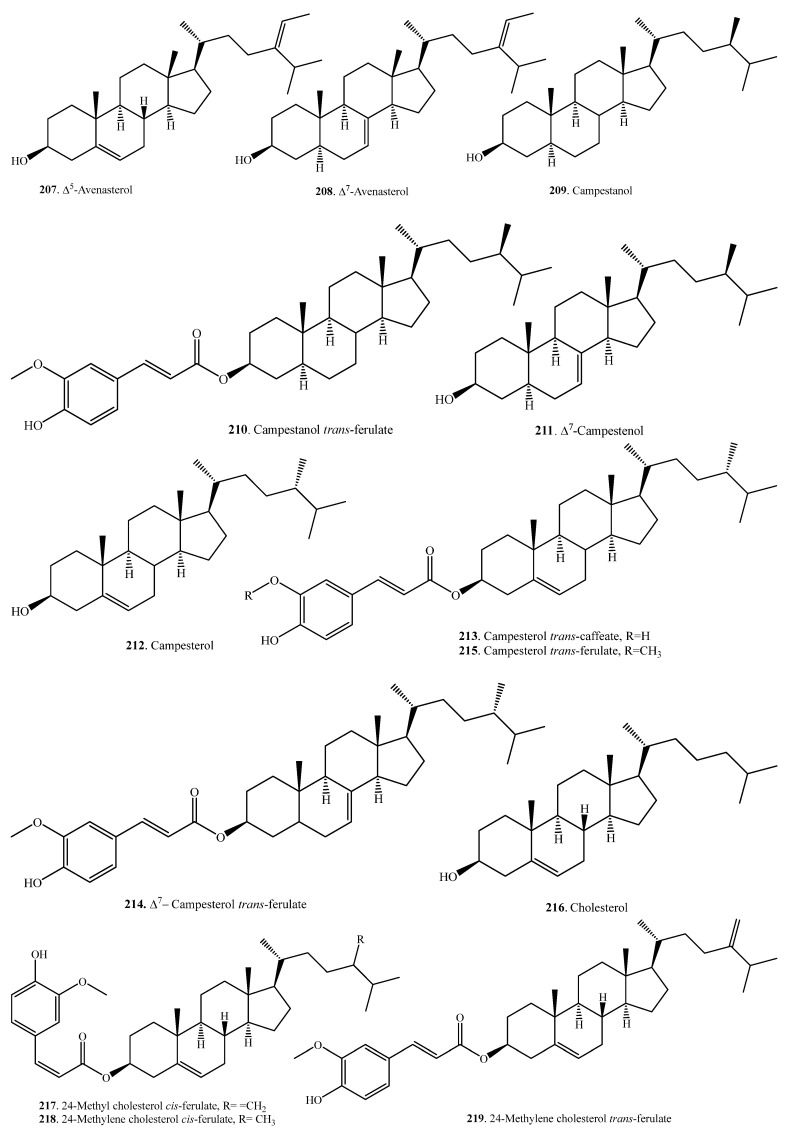
Structures of the steroids isolated from rice.

**Figure 8 molecules-23-03098-f008:**
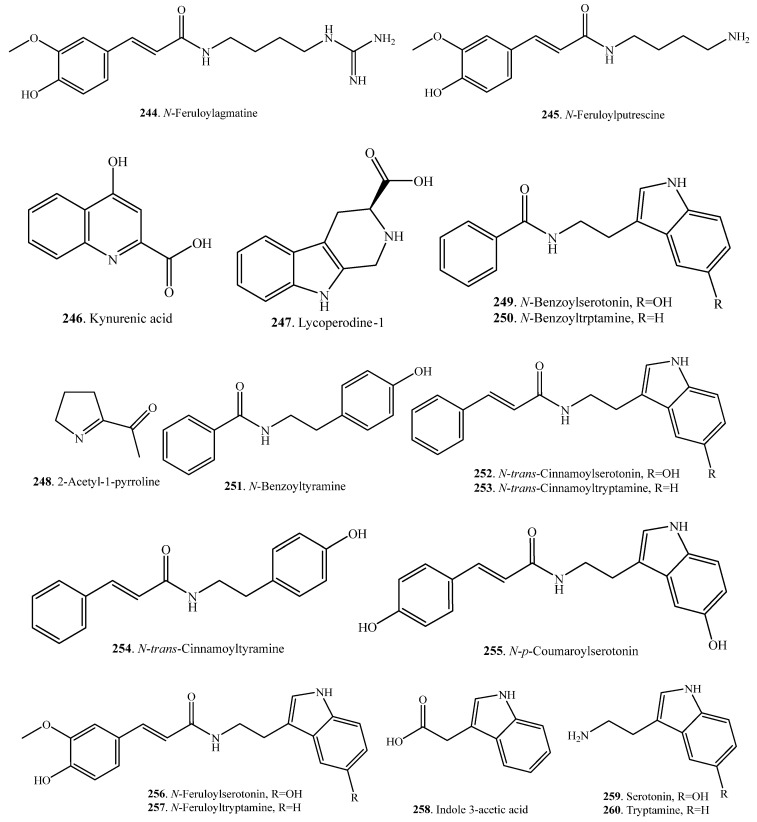
Structures of the alkaloids isolated from rice.

**Figure 9 molecules-23-03098-f009:**
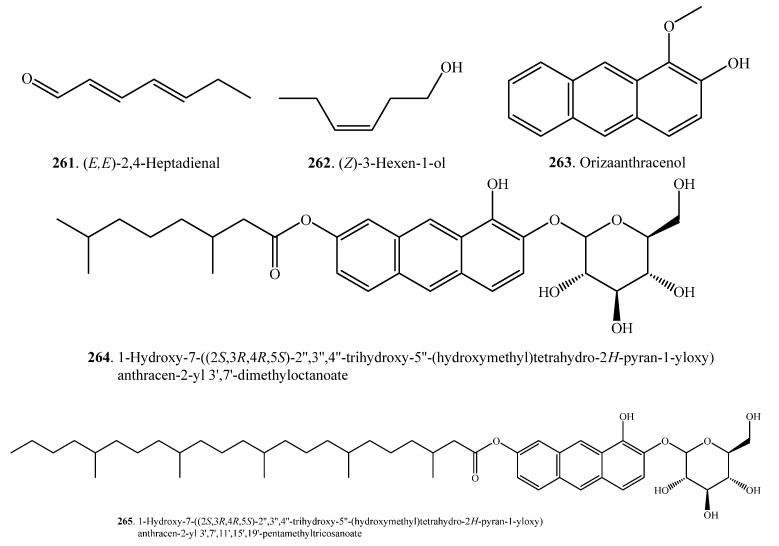
Structures of the other compounds isolated from rice.

**Figure 10 molecules-23-03098-f010:**
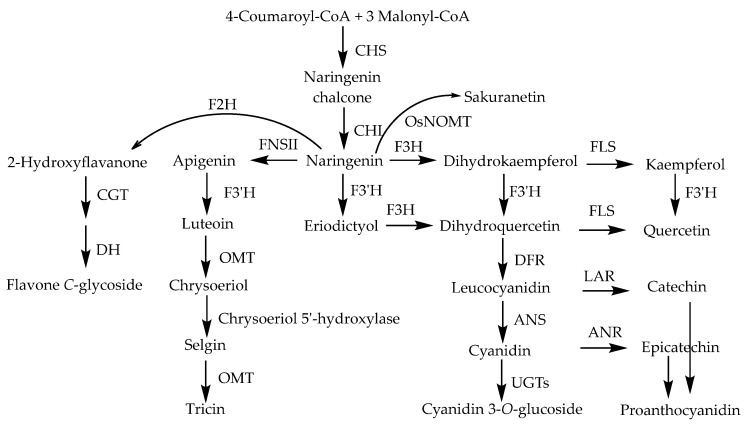
Proposed biosynthesis pathway of rice flavonoids [124]. Abbreviations: CHS, chalcone synthase; CHI, chalcone isomerase; F3H, flavanone 3-hydroxylase; FLS, flavonol synthase; F3′H, flavonoid 3′-hydroxylase; DFR, dihydroflavonol 4-reductase; ANS, anthocyanidin synthase; UGT, UDP-glucosyl transferase; LAR, leucoanthocyanidin reductase; ANR, anthocyanidin reductase; FNSII, flavone synthase II; OMT, *O*-methyltransferase; F2H, flavanone 2-hydroxylase; CGT, C-glucosyl transferase; and DH, dehydratase; OsNOMT, rice naringenin 7-*O*-methyltransferase.

**Figure 11 molecules-23-03098-f011:**
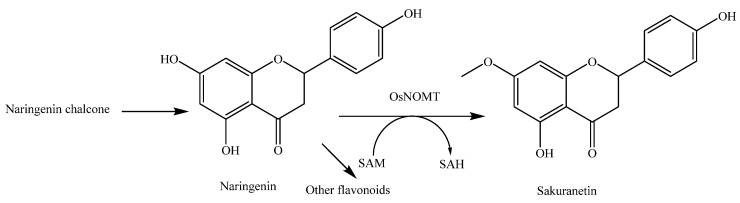
Biosynthesis pathway of sakuranetin in rice [9]. Abbreviations: SAM, *S*-adenosyl-l-methionine; SAH, *S*-adenosyl-l-homocysteine.

**Figure 12 molecules-23-03098-f012:**
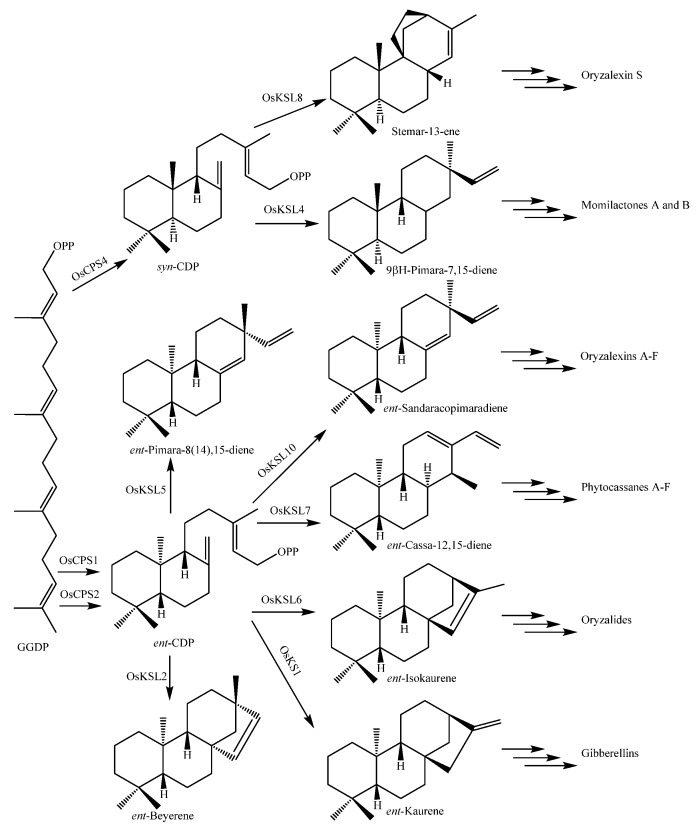
Biosynthesis pathways of the labdane-related diterpenoids in rice [141].

**Figure 13 molecules-23-03098-f013:**
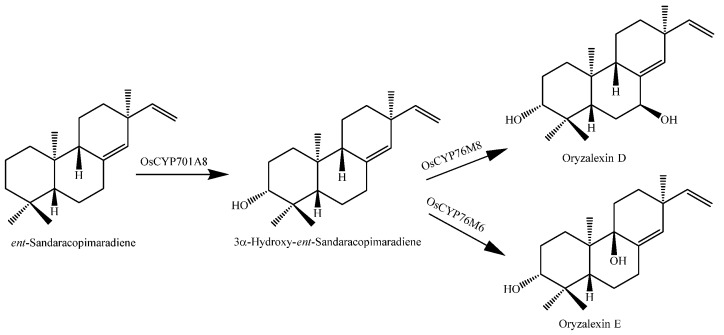
Biosynthesis pathways showing *ent*-sandaracopimaradiene dihydroxylation into oryzalexins D and E [144].

**Figure 14 molecules-23-03098-f014:**
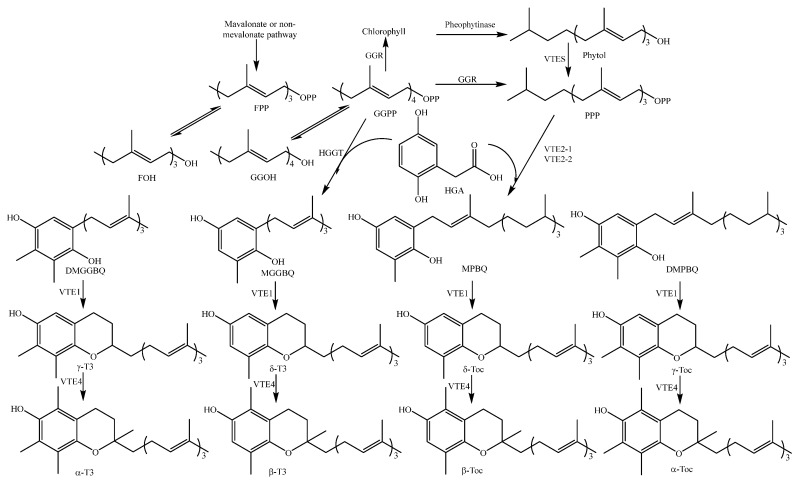
General biosynthetic pathway for vitamin E in plants [146]. Abbreviations: T3, tocotrienol; Toc, tocopherol; FOH, farnesol; GGOH, geranylgeraniol; FPP, farnesyl pyrophosphate; GGPP, geranylgeranyl pyrophosphate; PPP, phytyl pyrophosphate; HGA, homogentisic acid; MGGBQ, 2-methyl-6-geranylgeranylbenzoquinol; DMGGBQ, 2,3-dimethyl-6-geranylgeranylbenzoquinol; MPBQ, 2-methyl-6-phytylbenzoquinol; DMPBQ, 2,3-dimethyl-6-phytylbenzoquinol. The enzymes HGGT, GGR, VTE2-1,2, T3/Toc methyltransferase (VTE3 and VTE4), T3/Toc cyclase (VTE1), pheophytinase, and phytol kinase (VTE5) are believed to be involved in vitamin E biosynthesis.

**Figure 15 molecules-23-03098-f015:**
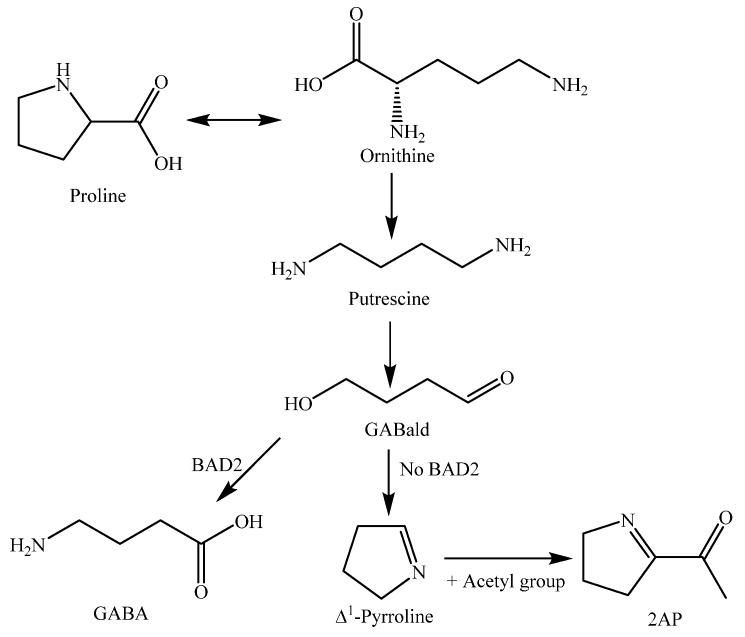
2-Acetyl-1-pyrroline (2AP) biosynthesis pathways in rice [149]. Abbreviations: GABald, γ-aminobutyraldehyde; BAD2, betaine aldehyde dehydrogenase; GABA, γ-aminobutyric acid.

**Figure 16 molecules-23-03098-f016:**
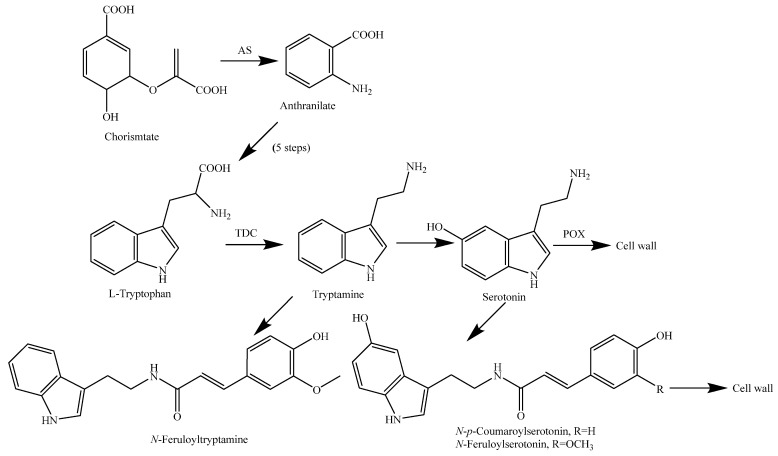
Alkaloid biosynthesis pathways in rice [147]. Abbreviations: AS, anthranilate synthase; TDC, tryptophan decarboxylase; POX, peroxidase.

**Table 1 molecules-23-03098-t001:** Phenolic acids and their biological activities.

Name	Rice Part Used for Isolation	Biological Activity and Function	Ref.
*p*-Hydroxybenzaldehyde (**1**)	Husk and bran	-	[24]
	Bran	Antioxidant activity	[31]
*p*-Hydroxybenzoic acid (**2**)	Root exudate	Allelopathic effect	[30]
	Husk and bran	-	[24]
*p*-Hydroxy methyl benzoate glucoside (**3**)	Bran	Antioxidant activity	[31]
*p*-Hydroxy phenyl acetaldehyde (**4**)	Husk and bran	-	[24]
*p*-Hydroxy phenyl acetic acid (**5**)	Husk and bran	-	[24]
2-Hydroxy 5-[(3*S*)-3-hydroxybutyl] phenyl β-d-glucoside (HHPG) (**6**)	Brans of purple rice	Inhibitory activity on tunicamycin-induced retinal damage	[32]
Caffeic acid (**7**)	Endosperm and bran/embryo of *indica* variety	Antioxidant activity	[20]
	Root exudate	Allelopathic effect	[30]
	Husk and bran	-	[24]
Methyl caffeate (**8**)	Bran	Antioxidant activity	[31]
Caffeoyl quinic acid methyl ester (**9**)	Grains of brown rice	-	[33]
Protocatechuic acid (**10**)	Endosperm and bran/embryo of *indica* variety	Antioxidant activity	[20]
Chlorogenic acid (**11**)	Endosperm and bran/embryo of *indica* variety	Antioxidant activity	[20]
Cinnamic acid (**12**)	Husk and bran	-	[24]
*o*-Coumaric acid (**13**)	Endosperm and bran/embryo of *indica* variety	Antioxidant activity	[20]
	Root exudate	Allelopathic effect	[30]
*m*-Coumaric acid (**14**)	Grains	-	[23]
*p*-Coumaric acid (**15**)	Grains	-	[23]
	Grains of brown rice	-	[33]
3-*O*-*p*-Coumaroyl quinic acid (**16**)	Grains of brown rice	-	[33]
	Leaves	-	[34]
*trans*-Ferulic acid (**17**)	Endosperm and bran/embryo of *indica* variety	Antioxidant activity	[20]
	Grains	-	[23]
	Black rice bran	Antioxidant activity	[25]
	Husk and bran	-	[24]
	Bran	Antioxidant activity	[31]
*trans*-Ferulic acid methyl ester (**18**)	Bran	Antioxidant activity	[31]
*cis*-Ferulic acid (**19**)	Bran	Antioxidant activity	[31]
*cis*-Ferulic acid methyl ester (**20**)	Bran	Antioxidant activity	[31]
1,3-*O*-Diferuloylglycerol (**21**)	Leaves	-	[34]
1-*O*-Feruloyl-β-d-glucose (**22**)	Leaves	-	[34]
3-*O*-Feruloylquinic acid (**23**)	Leaves	-	[34]
Gallic acid (**24**)	Endosperm and bran/embryo of *indica* variety	Antioxidant activity	[20]
	Husk and bran	-	[24]
*m*-Salicylic acid (**25**)	Grains of brown rice	-	[33]
Salicylic acid 2-*O*-β-d-glucopyranoside (**26**)	Leaves	-	[34]
Sinapic acid (**27**)	Grains	-	[23]
1-*O*-Sinapoyl-β-d-glucose (**28**)	Leaves	-	[34]
Syringaldehyde (**29**)	Grains of brown rice	-	[33]
Syringic acid (**30**)	Endosperm and bran/embryo of *indica* variety	Antioxidant activity	[20]
	Root exudate	Allelopathic effect	[30]
	Husk and bran	-	[24]
	Grains of brown rice	-	[33]
Vanillic aldehyde (**31**)	Bran	Antioxidant activity	[31]
Vanillic acid (**32**)	Root exudate	Allelopathic effect	[30]
	Husk and bran	-	[24]
	Grains of brown rice	-	[33]

**Table 2 molecules-23-03098-t002:** Flavonoids and their biological activities.

Name	Rice Part Used for Isolation	Biological Activity and Function	Ref.
Flavones			
Apigenin 6-*C*-α-l-arabinosyl-8-*C*-β-l-arabinoside (**33**)	Leaves	-	[53]
Chrysoeriol 7-*O*-rutinoside (**34**)	Grains of brown rice	-	[33]
Chrysoeriol 6-*C*-α-l-arabinosyl-8-*C*-β-l-arabinoside (**35**)	Leaves	-	[53]
5,7,4′-Trihydroxy-3′,5′-dimethoxyflavone (**36**)	Leaves	Allelopathic activity; antifungal activity	[54,55]
	Seedlings	Allelopathic activity	[41]
5,4′-Dihydroxy-3′,5′-dimethoxy-7-*O*-β-glucopyranosylflavone (**37**)	Seedlings	-	[41]
7,4′-Dihydroxy-3′,5′-dimethoxy-5-*O*-β-glucopyranosylflavone (**38**)	Seedlings	-	[41]
Isoscoparin 2-*O*-(6-(*E*)-feruloyl)-glucopyranoside (**39**)	Leaves	-	[53]
Isoscoparin 2″-*O*-(6‴-(*E*)-*p*-coumaroyl)-glucopyranoside (**40**)	Leaves	-	[53]
Isovitexin 2″-*O*-(6‴-(*E*)-feruloyl)-glucopyranoside (**41**)	Leaves	-	[53]
Isovitexin 2″-*O*-(6‴-(*E*)-*p*-coumaroyl)-glucopyranoside (**42**)	Leaves	-	[53]
Isoorientin 7,3′-dimethyl ether (**43**)	Leaves	-	[53]
luteolin 6-*C*-(2″-*O*-β-d-glucopyranosyl)-α-l-arabinoside (**44**)	Leaves	-	[53]
Schaftoside (**45**)	Leaves	Antifeedant activity	[35]
Isoschaftoside (**46**)	Leaves	Antifeedant activity	[35]
Swertisin (**47**)	Leaves	-	[53]
Tricin (**48**)	Bran	DPPH radical scavenging activity	[56]
Tricin 7-*O*-β-d-glucopyranoside (**49**)	Leaves	-	[53]
Tricin 5-*O*-β-d-glucopyranoside (**50**)	Leaves	-	[53]
Tricin 7-*O*-rutinoside (**51**)	Leaves	-	[53]
Tricin 7-*O*-neohesperidoside (**52**)	Leaves	-	[53]
Tricin 7-*O*-(2″-*O*-β-d-glucopyranosyl)-β-d-glucuronopyranoside (**53**)	Leaves	-	[53]
Tricin 7-*O*-(6″-*O*-malonyl)-β-d-glucopyranoside (**54**)	Leaves	-	[53]
Tricin 7-*O*-(6″-(*E*)-sinapoyl)-β-d-glucopyranoside (**55**)	Leaves	-	[53]
Tricin 4′-*O*-(*threo*-β-syringylglyceryl) ether 7″-*O*-β-d-glucopyranoside (**56**)	Leaves	-	[53]
Tricin 4′-*O*-(*erythro*-β-guaiacylglyceryl) ether (**57**)	Bran	DPPH radical scavenging activity	[56]
	Bran	Cytotoxicity and apoptosis induction in multiple tumor cells	[38]
Tricin 4′-*O*-(*threo*-β-guaiacylglyceryl) ether (**58**)	Bran	DPPH radical scavenging activity	[56]
	Bran	Cytotoxicity and apoptosis induction in multiple tumor cells	[38]
Tricin 4′-*O*-(*erythro*-β-guaiacylglyceryl) ether 7-*O*-β-d-glucopyranoside (**59**)	Leaves	-	[53]
Tricin 4′-*O*-(*threo*-β-guaiacylglyceryl) ether 7-*O*-β-d-glucopyranoside (**60**)	Leaves	-	[53]
Tricin 4′-*O*-(*erythro*-β-guaiacylglyceryl) ether 7″-*O*-β-d-glucopyranoside (**61**)	Leaves	-	[53]
Tricin 4′-*O*-(*threo*-β-guaiacylglyceryl) ether 7″-*O*-β-d-glucopyranoside (**62**)	Leaves	-	[53]
Tricin 4′-*O*-(*erythro*-β-guaiacylglyceryl) ether 9″-*O*-β-d-glucopyranoside (**63**)	Leaves	-	[53]
Tricin 4′-*O*-(*threo*-β-4-hydroxyphenylglyceryl) ether (64)	Leaves	-	[53]
Tricin 7-*O*-rutinoside (**65**)	Grains of brown rice	-	[33]
Flavonols			
Brassicin (**66**)	Grains of transgenic *japonica*	Radical scavenging activity	[39]
Brassicin-4′-*O*-β-d-glucopyranoside (**67**)	Grains of transgenic *japonica*	Radical scavenging activity	[39]
Isorhamnetin-4′-*O*-β-d-glucopyranoside (**68**)	Grains of transgenic *japonica*	Radical scavenging activity	[39]
Isorhamnetin-7-*O*-β-d-cellobioside (**69**)	Grains of transgenic *japonica*	Radical scavenging activity	[39]
Kaempferol (**70**)	Husk and bran	-	[24]
Myricetin (**71**)	Rice flour	-	[57]
Quercetin (**72**)	Rice flour	-	[57]
Quercetin 3-*O*-glucoside (**73**)	Rice flour	-	[57]
Quercetin 3-*O*-galactoside = Hyperoside (**74**)	Rice flour	-	[57]
Qucertin 3-O-rutinoside = Rutin (**75**)	Rice flour	-	[57]
Syringetin 3-*O*-β-d-glucopyranoside (**76**)	Leaves	-	[53]
Syringetin 3-*O*-rutinoside (**77**)	Leaves	-	[53]
Flavanones			
Hesperidin (**78**)	Rice flour	-	[57]
Naringenin (**79**)	Leaves	-	[42]
	-	Antifungal activity	[43,44]
	Rice flour	-	[57]
	Rice fungal pathogen	-	[44,45]
Naringenin 7-*O*-β-d-xylopyranoside (**80**)	Rice fungal pathogen	-	[45]
Sakuranetin (**81**)	Leaves	Antifungal activity	[43,44]
	Leaves	Antibacterial and antifungal activities	[58]
	Leaves	Anti-*Helicobacter pylori* activity	[49]
	-	Antileishmanial and antitrypanosomal activities	[50]
	-	Antioxidant activity	[42]
	-	Anti-inflammatory activity	[47]
	-	Anti-mutagenic activity	[48]
	-	Induction of adipogenesis in 3T3-L1 cells	[46]
	-	Induction of melanogenesis in B16BL6 melanoma cells	[51]
Sakuranetin 4′-*O*-β-d-xylopyranoside (**82**)	Rice fungal pathogen	-	[45]
Sternbin (**83**)	Rice fungal pathogen	-	[44]
Flavanonols			
3′-*O*-Methyltaxifolin (**84**)	Grains of transgenic *japonica*	Radical scavenging activity	[39]
3′-*O*-Methyltaxifolin-7-*O*-β-d-glucopyranoside (**85**)	Grains of transgenic *japonica*	Radical scavenging activity	[39]
3′-*O*-Methyltaxifolin-4′-*O*-β-d-glucopyranoside (**86**)	Grains of transgenic *japonica*	Radical scavenging activity	[39]
3′-*O*-Methyltaxifolin-5-*O*-β-d-glucopyranoside (**87**)	Grains of transgenic *japonica*	Radical scavenging activity	[39]
Flavanols			
Catechin (**88**)	Rice flour	-	[57]
Epicatechin (**89**)	Rice flour	-	[57]
Anthocyanins			
Cyanidin (**90**)	Bran	-	[59]
	Black rice kernels	Antioxidant activity	[52]
Cyanidin 3-*O*-gentiobioside (**91**)	Bran	-	[60]
	-	Inhibitory activity on tunicamycin-induced retinal damage	[32]
Cyanidin 3-*O*-glucoside (**92**)	Bran	-	[60]
		Inhibitory activity on tunicamycin-induced retinal damage	[32]
Cyanidin 3-*O*-rutinoside (**93**)	Kernels	-	[61]
Cyanidin 3-*O*-sambubioside (**94**)	Black rice kernels	Antioxidant activity	[52]
Cyanidin 3,5-*O*-diglucoside (**95**)	Kernels	-	[61]
Delphinidin (**96**)	Bran	-	[59]
Malvidin (**97**)	Bran	-	[59]
Pelargonidin (**98**)	Bran	-	[59]
Pelargonidin 3,5-*O*-diglucoside (**99**)	Pigmented rice	Antioxidant activity	[2]
Peonidin (**100**)	Black rice kernels	Antioxidant activity	[52]
Peonidin 3-*O*-glucoside (**101**)	Bran	-	[60]
	-	Inhibitory activity on tunicamycin-induced retinal damage	[32]
	Black rice kernels	Antioxidant activity	[52]

**Table 3 molecules-23-03098-t003:** Monoterpenoids and their biological activities.

Name	Rice Part Used for Isolation	Biological Activity and Function	Ref.
Camphene (**102**)	Bran	-	[65]
Camphor (**103**)	Bran	-	[65]
Carveol (**104**)	Bran	-	[65]
1,4-Cineol (**105**)	Bran	-	[65]
Fenchyl acetate (**106**)	Bran	-	[65]
(*S*)-Limonene (**107**)	Leaves	-	[66]
	Bran	-	[65]
	Seedlings	Antibacterial activity on *Xoo*	[63]
Linalool (**108**)	Leaves	-	[66]
	Leaves	Resistance induction to *Xoo*	[64]
*cis*-Linalool oxide (**109**)	Bran	-	[65]
*trans*-Linalool oxide (**110**)	Bran	-	[65]
Myrcene (**111**)	Seedlings	-	[63]
	Bran	-	[65]
*trans*-β-Ocimene (**112**)	Bran	-	[65]
α-Pinene (**113**)	Seedlings	-	[63]
β-Pinene (**114**)	Bran	-	[65]
Sabinene (**115**)	Seedlings	-	[63]
	Bran	-	[65]
α-Terpinene (**116**)	Seedlings	-	[63]
γ-Terpinene (**117**)	Leaves	Antibacterial activity on *Xoo*	[62]
Terpinen-4-ol (**118**)	Bran	-	[65]
α-Thujene (**119**)	Seedlings	-	[63]

**Table 4 molecules-23-03098-t004:** Sesquiterpenoids and their biological activities and functions.

Name	Rice Part Used for Isolation	Biological Activity and Function	Ref.
Abscisic acid (**120**)	Whole rice plant	Regulation of growth and development	[69]
(*Z*)-α-Bergamotene (**121**)	Leaves	-	[66]
β-Bisabolene (**122**)	Bran	-	[65]
(*E*)-γ-Bisabolene (**123**)	Leaves	-	[66]
α-Cadinene (**124**)	Leaves	-	[66]
β-Caryophyllene (**125**)	Leaves	-	[66]
	Bran	-	[65]
α-Copaene (**126**)	Leaves	-	[66]
	Bran	-	[65]
	Seedlings	-	[63]
α-Curcumene (**127**)	Leaves		[66]
γ-Curcumene (**128**)	Leaves		[66]
Cyclosativene (**129**)	Seedlings	-	[63]
α-Elemene (**130**)	Bran	-	[65]
β-Elemene (**131**)	Seedlings	-	[63]
(*E*)-β-Farnesene (**132**)	Leaves	-	[68]
Germacrene D (**133**)	Leaves		[66]
α-Gurjunene (**134**)	Bran	-	[65]
β-Gurjunene (**135**)	Leaves		[66]
α-Humulene (**136**)	Leaves		[66]
Italicene (**137**)	Leaves		[66]
γ-Muurolene (**138**)	Leaves		[66]
(*E*)-Nerolidol (**139**)	Leaves	Antibacterial activity against *Xoo*	[68]
7-*epi*-α-Selinene (**140**)	Bran	-	[65]
Valencene (**141**)	Leaves	-	[66]
Viridiflorene (**142**)	Leaves	-	[66]
α-Ylangene (**143**)	Bran	-	[65]
α-Zingiberene (**144**)	Leaves	-	[66]

**Table 5 molecules-23-03098-t005:** Diterpenoids and their biological activities.

Name	Rice Part Used for Isolation	Biological Activity and Function	Ref.
Phytohormone gibberellins			
Gibberellin A_1_ (**145**)	Whole plant	Growth-promoting activity	[70]
Gibberellin A_4_ (**146**)	Whole plant	Growth-promoting activity	[70]
Gibberellin A_19_ (**147**)	Whole plant	Growth-promoting activity	[70]
Pimaradiene-type diterpenoids			
Momilactone A (**148**)	Coleoptiles	Antifungal activity	[71]
	Bran	Growth inhibitory activity on rice roots	[90]
	Bran	Inhibitory activities on seed germination and growth of barnyard grass	[93]
	Root exudates	Allelopathy effect	[91]
Momilactone B (**149**)	Coleoptiles	Antifungal activity	[71]
	Seedlings	Growth inhibitory activity on rice roots	[90,92]
	Seedlings	Allelopathic effects	[40]
	Bran	Inhibitory activities on seed germination and growth of barnyard grass	[93]
	Root exudates	Allelopathy effect	[91]
Momilactone C (**150**)	Bran	Weak growth inhibitory activity	[94]
Momilactone D (**151**)	Roots	-	[95]
Momilactone E (**152**)	Roots	-	[95]
9β-Pimara-7,15-diene-3β,6β,19-triol (**153**)	Leaves	Weak antifungal activity	[72]
*ent*-Sandaracopimaradiene-type diterpenoids			
*ent*-15,16-Epoxy-2,3-dihydroxy- kaurane (**154**)	Leaves of a bacterial leaf blight-resistant cultivar	Antibacterial activity	[85]
*ent*-2,3,15-Trihydroxy- kaurane (**155**)	Leaves of a bacterial leaf blight-resistant cultivar	Antibacterial activity	[85]
*ent*-15,16-Epoxy-kauran-3-one (**156**)	Leaves of a bacterial leaf blight-resistant cultivar	Antibacterial activity	[85]
*ent*-15,16-Epoxy-kauran-2,3-dione = Oryzadione (**157**)	Leaves of a bacterial leaf blight-resistant cultivar	Antibacterial activity	[86]
*ent*-15,16-Epoxy-3β-hydroxy-kauran-2-one (**158**)	Leaves of a bacterial leaf blight-resistant cultivar	Antibacterial activity	[86]
*ent*-15,16-Epoxy-3-oxa-kauran-2-one (**159**)	Leaves of a bacterial leaf blight-resistant cultivar	Antibacterial activity	[86]
*ent*-15,16-Epoxy-3β-myristoyloxy-kauran-2-one (**160**)	Leaves of a bacterial leaf blight-resistant cultivar	Antibacterial activity	[86]
*ent*-15,16-Epoxy-3α-palmitoyloxy-kauran-2-one (**161**)	Leaves of a bacterial leaf blight-resistant cultivar	Antibacterial activity	[86]
*ent*-15,16-Epoxy-3β-palmitoyloxy-kauran-2-one (**162**)	Leaves of a bacterial leaf blight-resistant cultivar	Antibacterial activity	[86]
Oryzalexin A (**163**)	Leaves	Inhibitory activity on spore germination and germ tube growth of *O**chrobactrum* *oryzae*	[73,76]
	Roots	-	[95]
Oryzalexin B (**164**)	Leaves	Inhibitory activity on spore germination and germ tube growth of *O. oryzae*	[75,76]
Oryzalexin C (**165**)	Leaves	Inhibitory activity on spore germination and germ tube growth of *O.* *oryzae*	[75,76]
Oryzalexin D(**166**)	Leaves	Inhibitory activity on spore germination of *Magnaporthe Oryzae*	[77]
Oryzalexin E (**167**)	Leaves	Inhibitory activity on spore germination of *M. Oryzae*	[78]
Oyzalexin F (**168**)	Leaves	Antimicrobial activity	[79]
Oryzalic acid A (**169**)	Leaves of a bacterial leaf blight-resistant cultivar	Antibacterial activity	[88]
Oryzalic acid B = *ent*-15-Hydroxy-2,3-secokauren- 2,3-dioic acid (**170**)	Leaves of a bacterial leaf blight-resistant cultivar	Antibacterial activity	[85]
Oryzalide A = *ent*-15,16-Epoxy-1α-hydroxy-2-oxa-kauran-3-one (**171**)	Leaves of a bacterial leaf blight-resistant cultivar	Antibacterial activity	[87,88]
Oryzalide B (**172**)	Leaves of a bacterial leaf blight-resistant cultivar	Antibacterial activity	[88]
Sandaracopimaradien-3-one (**173**)	Roots	-	[95]
Stemarene-type diterpenoids			
Oryzalexin S (**174**)	Leaves	Antifungal activity	[96,97]
Stemar-13-en-2α-ol (**175**)	Leaves	Weak antifungal activity	[72]
*ent*-Cassadiene-type diterpenoids			
Phytocassane A (**176**)	Leaves infected with *M. oryzae*; stems infected with *Rhizoctonia Solani*	Antifungal activity	[80]
Phytocassane B (**177**)	Leaves infected with *M. oryzae*; stems infected with *R. Solani*	Antifungal activity	[80]
Phytocassane C (**178**)	Leaves infected with *M. oryzae*; stems infected with *R. Solani*	Antifungal activity	[80]
Phytocassane D (**179**)	Leaves infected with *M. oryzae*; stems infected with *R. Solani*	Antifungal activity	[80]
Phytocassane E (**180**)	Cultured rice cells	Inhibition activity on spore germination and germ tube growth of *M. oryzae*	[81]
Phytocassane F (**181**)	Leaves	Antifungal activity	[72]
Casbene-type diterpenoids			
5-Deoxo-*ent*-10-oxodepressin (**182**)	Leaves	Antifungal activity	[83]
5-Dihydro-*ent*-10-oxodepressin (**183**)	Leaves	Antifungal activity	[83]
*ent*-10-Oxodepressin (**184**)	Leaves	Antifungal activity	[84]

**Table 6 molecules-23-03098-t006:** Triterpenoids and their biological activities.

Name	Rice Part Used for Isolation	Biological Activity and Function	Ref.
Citrostadienol (**185**)	Bran	-	[105]
	Bran	Anti-inflammatory activity	[104]
Citrostadienol *cis*-ferulate (**186**)	Bran	Anti-inflammatory activity	[104]
Citrostadienol *trans*-ferulate (**187**)	Bran	Anti-inflammatory activity	[104]
(24*S*)-Cycloart-25-ene-3β,24-diol-3β-*trans*-ferulate (**188**)	Bran	Moderate cytotoxic activity	[99]
(24*R*)-Cycloart-25-ene-3β,24-diol-3β-*trans*-ferulate (**189**)	Bran	Moderate cytotoxic activity	[99]
Cycloart-23*Z*-ene-3β,25-diol-3β-*trans*-ferulate (**190**)	Bran	Moderate cytotoxic activity	[99]
Cycloartenol (**191**)	Bran	-	[105]
	Bran	Lowering postpradial hyperglyceimia	[103]
Cycloartenol *trans*-caffeate (**192**)	Seeds	-	[106]
Cycloartenol *cis*-ferulate (**193**)	Bran	-	[98]
Cycloartanol *trans*-ferulate (**194**)	Bran	-	[98]
	Bran	Moderate cytotoxic activity	[99]
24-Methylene cycloartanol (**195**)	Bran of black non-glutinous rice	Anti-cancer activity	[107]
	Bran	Lowering postpradial hyperglyceimia	[103]
24-Methylene cycloartanol *cis*-ferulate (**196**)	Bran	-	[98]
	Bran	Anti-inflammatory activity	[104]
24-Methylene cycloartanol *trans*-ferulate (**197**)	Bran	-	[98]
	Bran	Moderate cytotoxic activity	[99]
Cycloeucalenol (**198**)	Bran of black non-glutinous rice	Anti-cancer activity	[107]
Cycloeucalenol *cis*-ferulate (**199**)	Bran	Antioxidant activity	[31]
Cycloeucalenol *trans*-ferulate (**200**)	Bran	-	[98]
	Bran	Anti-inflammatory activity	[104]
	Bran	Antioxidant activity	[31]
Gramisterol (**201**)	Bran of black non-glutinous rice	Anti-cancer activity	[107]
Gramisterol *cis*-ferulate (**202**)	Bran	Anti-inflammatory activity	[104]
Gramisterol *trans*-ferulate (**203**)	Bran	Anti-inflammatory activity	[104]
Lanast-7,9(11)-dien-3α,15α-diol-3α-d-glucofuranoside (**204**)	Hulls	Herbicidal activity	[108]
Lupeol (**205**)	Bran of black non-glutinous rice	Anti-cancer activity	[107]
Lupenone (**206**)	Bran of black non-glutinous rice	Anti-cancer activity	[107]

**Table 7 molecules-23-03098-t007:** Steroids and their biological activities.

Name	Rice Part Used for Isolation	Biological Activity and Function	Ref.
∆^5^-Avenasterol (**207**)	Germinating seeds	-	[106]
∆^7^-Avenasterol (**208**)	Germinating seeds	-	[106]
Campestanol (**209**)	Germinating seeds	-	[106]
Campestanol *trans*-ferulate (**210**)	Bran	-	[98]
∆^7^-Campestenol (**211**)	Germinating seeds	-	[106]
Campesterol (**212**)	Bran	-	[107]
	Seedlings	Drought stress tolerance	[109]
Campesterol *trans*-caffeate (**213**)	Bran	-	[98]
∆^7^-Campesterol *trans*-ferulate (**214**)	Bran	-	[99]
Campesterol *trans*-ferulate (**215**)	Bran	-	[99]
Cholesterol (**216**)	Germinating seeds	-	[106]
24-Methyl cholesterol *cis*-ferulate (**217**)	Bran	Anti-inflammatory activity	[105]
24-Methylene cholesterol *cis*-ferulate (**218**)	Bran	Anti-inflammatory activity	[104]
24-Methylene cholesterol *trans*-ferulate (**219**)	Bran	-	[98]
	Bran	Anti-inflammatory activity	[104]
24-Methylene ergosta-5-en-3β-ol (**220**)	Bran	-	[107]
24-Methylene ergosta-7-en-3β-ol (**221**)	Bran	-	[107]
Fucosterol (**222**)	Bran	-	[107]
Schleicheol 2 (**223**)	Bran	-	[110]
Sitostanol (**224**)	Germinating seeds	-	[106]
Sitosterol = β-Sitosterol (**225**)	Bran	-	[105,107]
	Seedlings	Drought stress tolerance	[109]
7α-Hydroxy sitosterol (**226**)	Bran	-	[110]
7β-Hydroxy sitosterol (**227**)	Bran	-	[110]
Sitosterol *cis*-ferulate (**228**)	Bran	-	[98]
	Bran	Anti-inflammatory activity	[104]
Sitosterol *trans*-ferulate (**229**)	Bran	-	[98]
∆^7^-Sitosterol *trans*-ferulate (**230**)	Bran	-	[98]
d-Glucopyranosyl-(β1→4)-d-glucopyranosyl-(β1→3′)-β-sitosterol (**231**)	Bran (Hulls)	-	[111]
d-Glucopyranosyl-(β1→3)-d-glucopyranosyl-(β1→3′)-β-sitosterol (**232**)	Bran (Hulls)	-	[111]
d-Glucopyranosyl-(β1→4)-d-glucopyranosyl-(β1→4)-d-glucopyranosyl-(β1→3′)-β-sitosterol (**233**)	Bran (Hulls)	-	[111]
Cellotetraosylsitosterol (**234**)	Bran	-	[112]
Cellopentaosylsitosterol (**235**)	Bran	-	[112]
Stigmastanol *cis*-ferulate (**236**)	Bran	Anti-inflammatory activity	[104]
Stigmastanol *trans*-ferulate (**237**)	Bran		[98]
	Bran	Anti-inflammatory activity	[104]
Stigmastanol-3β-*p*-butanoxy dihydrocoumaroate (**238**)	Hulls	Weak herbicidal activity	[108]
Stigmastanol-3β-*p*-glyceroxy dihydrocoumaroate (**239**)	Hulls	-	[108]
∆^7^-Stigmastenol (**240**)	Germinating seeds	-	[106]
Stigmasterol (**241**)	Bran	-	[105,107]
	Seedlings	Drought stress tolerance	[109]
Stigmasterol *cis*-ferulate (**242**)	Bran	Anti-inflammatory activity	[104]
Stigmasterol *trans*-ferulate (**243**)	Bran	-	[98]
	Bran	Anti-inflammatory activity	[104]

**Table 8 molecules-23-03098-t008:** Alkaloids and their biological activities.

Name	Rice Part Used for Isolation	Biological Activity and function	Ref.
*N*-Feruloylagmatine (**244**)	Leaves	Antimicrobial activity	[116]
*N*-Feruloylputrescine (**245**)	Leaves	Antimicrobial activity	[116]
Kynurenic acid (**246**)	Leaves	-	[34]
Lycoperodine-1 (**247**)	Leaves	-	[34]
2-Acetyl-1-pyrroline (**248**)	Grains	-	[113]
*N*-Benzoylserotonin (**249**)	Leaves	Antimicrobial activity	[116]
*N*-Benzoyltryptamine (**250**)	Leaves	Antimicrobial activity	[116]
	Leaves	Antibacterial activity	[58]
*N*-Benzoyltyramine (**251**)	Leaves	Antimicrobial activity	[116]
*N-trans*-Cinnamoylserotonin (**252**)	Leaves	Antimicrobial activity	[116]
*N-trans*-Cinnamoyltryptamine (**253**)	Leaves	Antimicrobial activity	[116]
	Leaves	Antibacterial activity	[58]
*N-trans*-Cinnamoyltyramine (**254**)	Whole rice plant	Allelopathic activity; antifungal activity	[117]
	Leaves	Antibacterial activity	[58]
*N-p*-Coumaroylserotonin (**255**)	Leaves	Antimicrobial activity	[116]
	Leaves	Antibacterial activity	[58]
*N*-Feruloylserotonin (**256**)	Leaves	Antimicrobial activity	[116]
*N*-Feruloyltryptamine (**257**)	Leaves	-	[118]
Indole 3-acetic acid (**258**)	Whole rice plant	Regulation on growth and development	[119]
Serotonin = 5-Hydroxytryptamine (**259**)	Leaves	-	[118]
Tryptamine (**260**)	Leaves	-	[118]

**Table 9 molecules-23-03098-t009:** Other metabolites and their biological activities.

Name	Rice Part Used for Isolation	Biological Activity and Function	Ref.
(*E*,*E*)-2,4-Heptadienal (**261**)	Whole phants	Antibacterial and antifungal activities, toxic to rice plants	[120]
(*Z*)-3-Hexen-1-ol (**262**)	Leaves	-	[66]
Orizaanthracenol = 1-Methoxyanthracen-2-ol (**263**)	Hulls	Strong inhibitory activity in seed germination of radish	[121]
1-Hydroxy-7-((2S,3R,4R,5S)-2″,3″,4″-trihydroxy-5″-(hydroxymethyl)tetrahydro-2H-pyran-1-yloxy)anthracen-2-yl 3′,7′-dimethyloctanoate (**264**)	Hulls	Weak inhibitory activity in seed germination of radish	[121]
1-Hydroxy-7-((2S,3R,4R,5S)-2″,3″,4″-trihydroxy-5″-(hydroxymethyl)tetrahydro-2H-pyran-1-yloxy)anthracen-2-yl 3′,7′,11′,15′,19′-pentamethyltricosanoate (**265**)	Hulls	Weak inhibitory activity in seed germination of radish	[121]
(5*S*)-5-(Acetyloxy)-3-(1-methylenthyl)-2-cyclohexen-1-one = 3-Isopropyl-5-acetoxycyclohexene-2-one-1 (**266**)	Leaves	Allelopathic activity	[55]
	Seedlings	Allelopathic effects	[40]
*cis*-12-oxo-Phytodienoic acid (**267**)	Whole plants	Inducible anti-insect activity	[122]
1-Phenyl-2-hydroxy-3,7-dimethyl-11-aldehydic-tetradecane-2β-d-glucopyranoside (**268**)	Hulls	Herbicidal activity	[108]
α-Tocopherol (**269**)	Bran	Antioxidative, antihypercholesterolemic, anticancer, neuroprotective activities	[123]
β-Tocopherol (**270**)	Bran	Antioxidative, antihypercholesterolemic, anticancer, neuroprotective activities	[123]
γ-Tocopherol (**271**)	Bran	Antioxidative, antihypercholesterolemic, anticancer, neuroprotective activities	[123]
δ-Tocopherol (**272**)	Bran	Antioxidative, antihypercholesterolemic, anticancer, neuroprotective activities	[123]
α-Tocotrienol (**273**)	Bran	Antioxidative, antihypercholesterolemic, anticancer, neuroprotective activities	[123]
β-Tocotrienol (**274**)	Bran	Antioxidative, antihypercholesterolemic, anticancer, neuroprotective activities	[123]
γ-Tocotrienol (**275**)	Bran	Antioxidative, antihypercholesterolemic, anticancer, neuroprotective activities	[123]
δ-Tocotrienol (**276**)	Bran	Antioxidative, antihypercholesterolemic, anticancer, neuroprotective activities	[123]

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
