# Peer review of "Rice Secondary Metabolites: Structures, Roles, Biosynthesis, and Metabolic Regulation"

_molecules, 2018, doi:10.3390/molecules23123098_

Round 1

Reviewer 1 Report

The manuscript “Rice Secondary Metabolites: Structures, Roles, Biosynthesis and Metabolic Regulations” summarized the structures, biological functions and activities, biosynthesis, and metabolic regulations of rice secondary metabolites. The review is very interesting, the writing style and the use of English are accurate. The manuscript is well organized, conclusions are clear and reference list is complete. I have no specific scientific comment but it is necessary to correct the manuscript in few points before being considered for publication:

- in the Introduction: it could be useful to introduce a small paragraph about the relation of natural bioactive compounds, their antioxidant profile, and their beneficial effect for human health. In this sense I suggest to read and cite: DOI: 10.1021/acs.jafc.6b00857, DOI: 10.1111/1750-3841.14284.

- please, check all the abbreviations used in the text. The abbreviations have to be defined in parentheses the first time they appear in the text.

- in all the presented tables it could be useful to separate the in vitro from the in vivo studies.

- in my opinion tables 3 and 4 are not necessary and can be removed.

Author Response

General comments:

The manuscript “Rice Secondary Metabolites: Structures, Roles, Biosynthesis and Metabolic Regulations” summarized the structures, biological functions and activities, biosynthesis, and metabolic regulations of rice secondary metabolites. The review is very interesting, the writing style and the use of English are accurate. The manuscript is well organized, conclusions are clear and reference list is complete. I have no specific scientific comment but it is necessary to correct the manuscript in few points before being considered for publication:

Comment 1:

-in the Introduction: it could be useful to introduce a small paragraph about the relation of natural bioactive compounds, their antioxidant profile, and their beneficial effect for human health. In this sense I suggest to read and cite: DOI: 10.1021/acs.jafc.6b00857, DOI: 10.1111/1750-3841.14284.

Response: Thanks for the suggestions. We added a few sentences about the relation of natural bioactive compounds, their antioxidant profile, and their beneficial effect for human health in the section “Introduction” Both two mentioned papers were cited in the text. The sentences “…, which are implicated in various health-promoting and disease preventive effects. ….. Some metabolites such as phenolic acids and flavonoids are also distributed in other plant species [5,6]” were added in this section.

Comment 2:

-please, check all the abbreviations used in the text. The abbreviations have to be defined in parentheses the first time they appear in the text.

Response: All the abbreviations used in the text were checked once more.

Comment 3

-in all the presented tables it could be useful to separate the in vitro from the in vivo studies.

Response: It is a good suggestion. However, it is very difficult to separate the in vitro results from the in vivo. Many of them belong to the in vivo studies.

Comment 4

-in my opinion tables 3 and 4 are not necessary and can be removed.

Response: There are 18 compounds in Table 3, and 25 compounds in Table 4. If Tables 3 and 4 were removed, these compounds could not be well described in the text though their biological activities were not determined. So, we still kept these two tables.

Reviewer 2 Report

The work presented for review describes in a complex way the metabolites present in rice and their biosynthetic pathways. In the reviewer's opinion, the work is too extensive. One of the proposals is to divide the work into two parts and publish it as 2 separate articles. The first relates to compounds found in rice, while the second relates to the biosynthesis and metabolic transformation of nineteenth compounds.  The length of work is also influenced by the number of chemical compound patterns found in the work. It is possible to avoid presenting the structural forms of all compounds by presenting forms of basic phenolic acids and placing them in the table. In the opinion of the reviewer, the structure of only those compounds characteristic of rice is important, which distinguishes it from other grains.

Author Response

Comment:

The work presented for review describes in a complex way the metabolites present in rice and their biosynthetic pathways. In the reviewer's opinion, the work is too extensive. One of the proposals is to divide the work into two parts and publish it as 2 separate articles. The first relates to compounds found in rice, while the second relates to the biosynthesis and metabolic transformation of nineteenth compounds. The length of work is also influenced by the number of chemical compound patterns found in the work. It is possible to avoid presenting the structural forms of all compounds by presenting forms of basic phenolic acids and placing them in the table. In the opinion of the reviewer, the structure of only those compounds characteristic of rice is important, which distinguishes it from other grains.

Response: Thanks for the kind comments and suggestions. Please forgive us for not dividing the review into two parts as we think that the description of the metabolites is the basis of biosynthesis and biosynthetic regulation. The structural characteristics and biosynthesis pathways of some compounds such as diterpenoid phytoalexins are unique to rice. We have discussed them in the section “Conclusions and Future Perspectives”.